# Entropy-aware Span-Constrained Optimal Transport for Robust Cross-Tokenizer Knowledge Distillation

**Zhi-Ping Liu** [* 1 2]  **Simiao Li** [* 3]  **Wei Li** [† 3]  **Hanting Chen** [3]  **Jie Hu** [3]  **Hua-Lei Yin** [‡ 2 1]  **Xinghao Chen** [‡ 3]

## Abstract

Existing Cross-Tokenizer Knowledge Distillation (CTKD) methods can fail to outperform simple supervised fine-tuning when vocabulary overlap is low due to severe alignment noise. We identify this phenomenon as the **"Low-Overlap negative transfer regime"**. To overcome this, we propose **Entropy-aware Span-Constrained Optimal Transport (E-SCOT)**, a robust framework that treats distillation as a sparse transport problem built upon a vocabulary-agnostic ground metric. Unlike prior OT approaches that incur quadratic costs via dense sequence-level optimization, E-SCOT employs span-anchored lexical alignment to construct a deterministic, locality-preserving support set in linear time with respect to sequence length. Furthermore, we introduce Rényi-entropy adaptive reweighting to dynamically concentrate the distillation budget on informative positions exhibiting significant uncertainty-profile gaps. Extensive experiments demonstrate that E-SCOT achieves state-of-the-art performance across diverse model families, effectively eliminating negative transfer even in challenging low-overlap scenarios.

## 1. Introduction

Large Language Models (LLMs) have demonstrated remarkable capabilities across a wide range of natural language processing tasks. However, the immense computational cost and memory footprint of these models hinder their deploy-

---

[*]Equal contribution  [†]Project leader.  [‡]Corresponding author. [1]National Laboratory of Solid State Microstructures and School of Physics, Nanjing University, Nanjing, China [2]School of Physics and Key Laboratory of Quantum State Construction and Manipulation (Ministry of Education), Renmin University of China, Beijing, China [3]Huawei Foundation Model Dept.. Correspondence to: Hua-Lei Yin <hlyin@ruc.edu.cn>, Xinghao Chen <xinghao.chen@huawei.com>.

*Proceedings of the 43rd International Conference on Machine Learning*, Seoul, South Korea. PMLR 306, 2026. Copyright 2026 by the author(s).

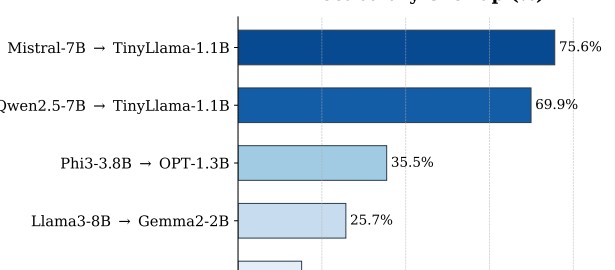

*Figure 1.* **Vocabulary Overlap Statistics.** Vocabulary overlap ratios between teacher and student models from diverse model families.

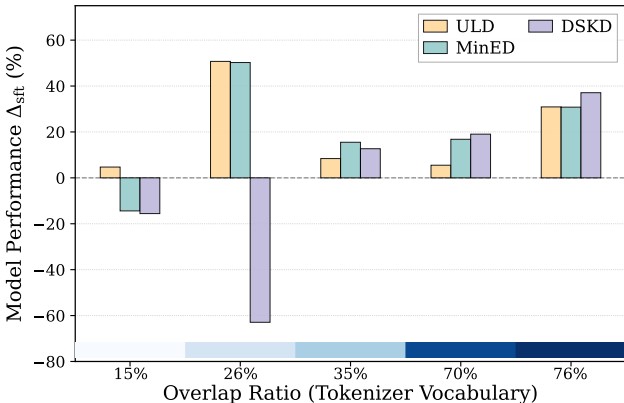

*Figure 2.* **The Low-Overlap negative transfer regime.** Performance comparison of representative CTKD methods (MinED, DSKD, ULD) relative to SFT across varying vocabulary overlap ratios. $\Delta_{\text{SFT}} < 0$ indicates negative transfer, where the distillation method underperforms the SFT baseline.

ment in resource-constrained environments. Knowledge Distillation (KD) (Hinton et al., 2015) has emerged as a standard paradigm to address this by transferring knowledge from a large teacher model to a smaller student model. While standard KD is effective when both models share the same tokenizer, practical scenarios increasingly demand **Cross-Tokenizer Knowledge Distillation (CTKD)** (Wan et al., 2024; Zhang et al., 2024b; Boizard et al., 2025; Chen et al., 2025), where the teacher and student utilize disparate vocabularies (e.g., distilling a Deepseek (Liu et al., 2024) teacher into a Qwen (Yang et al., 2024) student).

The core challenge in CTKD lies in the mismatch between token spaces induced by different teacher and student tokenizers. Standard KD relies on Kullback-Leibler (KL) Divergence (Gu et al., 2024; Ko et al., 2024; Wu et al., 2025) to minimize the discrepancy between teacher and student logit distributions. To apply this in cross-tokenizer settings, two fundamental alignment problems must be solved: (1) *Sequence Alignment*, ensuring the teacher provides supervision at the correct temporal step, and (2) *Vocabulary Alignment*, mapping the teacher's probability distribution to the student's vocabulary space.

Existing approaches have attempted to bridge these gaps. Methods like MinED (Wan et al., 2024) and DSKD (Zhang et al., 2024b) retain the use of divergence-based objectives by bridging the vocabulary gap through explicit token mapping or hidden-state projection. However, these vocabulary alignment mechanisms inevitably introduce noise alongside valid supervised signals. We can expect that the severity of such alignment errors increases with the increasing degree of vocabulary mismatch between the teacher and the student. This is particularly detrimental given that divergence-based objectives are notoriously sensitive to noise in distribution (Li et al., 2025), often leading to the suboptimal transfer (or even negative transfer) when alignment errors accumulate. We quantify the vocabulary mismatch with an intuitive metric: the vocabulary overlap ratio between teacher and student models. Vocabulary overlap statistics are shown in Figure 1. As illustrated in Figure 2, we observe a distinct performance degradation in existing CTKD methods as the vocabulary overlap ratio decreases. As the vocabulary overlap decreases, the distillation noise substantially undermines the effectiveness of knowledge transfer, leading to a **Low-Overlap negative transfer regime** where the student underperforms a simple Supervised Fine-Tuning (SFT) baseline (i.e., $\Delta_{\text{SFT}} < 0$).

Alternatively, methods like ULD (Boizard et al., 2025) attempt to bypass explicit vocabulary alignment by leveraging OT theory. While promising, ULD simplifies the OT problem to a strictly token-level calculation, ignoring the structural dependencies inherent in language sequences. This token-level treatment limits its performance. Although MultiLevelOT (Cui et al., 2025) extends ULD by jointly optimizing token-level cost matrices and employing Sinkhorn approximations for sequence-level cost matrices, this global approach incurs substantial computational overhead and lacks the explicit locality and monotonicity essential for efficient alignment. Moreover, these methods typically apply uniform supervision intensity across the sequence, neglecting that the learning difficulty varies significantly across tokens, especially when valid signals are diluted by tokenizer mismatches.

To address misalignment and adaptive signal-allocation issues in cross-tokenizer distillation, we propose **Entropy-aware Span-Constrained Optimal Transport (E-SCOT)**. E-SCOT adopts an OT-inspired formulation for cross-tokenizer distillation, aggregating pairwise teacher–student token-position discrepancies over a sparse, locality-preserving matching structure. Rather than solving a dense sequence-level OT optimization during training (Cui et al., 2025), E-SCOT uses the shared raw text as a reference to construct an efficient sequence alignment, and then performs distillation only on the resulting sparse interactions.

Concretely, E-SCOT consists of three components. **(i) Vocabulary-Agnostic Ground Metric.** To resolve vocabulary mismatch, we adopt the shape-based Wasserstein distance from ULD (Boizard et al., 2025), which compares the sorted confidence profiles of distributions. This allows us to quantify token-level discrepancies robustly without requiring shared indices or learnable projections. **(ii) Span-anchored lexical alignment.** Instead of solving expensive dense transport problems, we construct a sparse *support set* via character-span overlaps and apply a uniform local matching prior. This enforces strict linguistic locality and reduces the alignment complexity to linear time of token sequence length, effectively avoiding the quadratic overhead of iterative sequence-level OT solvers. **(iii) Rényi-Entropy adaptive reweighting.** Finally, we introduce a Rényi-entropy (Rényi, 1961) based token reweighting scheme to construct the cost matrix and allocate more distillation budget to informative positions where the teacher and student exhibit larger uncertainty-profile gaps.

Our contributions are summarized as follows:

- We identify the "Low-Overlap negative transfer regime" in CTKD, showing that vocabulary mismatch can lead to negative transfer where students underperform SFT baselines.

- We propose **E-SCOT**, an OT-inspired cross-tokenizer distillation framework that builds on a ULD-based vocabulary-agnostic ground metric and introduces two key mechanisms: span-anchored sparse support for locality-preserving alignment and Rényi-entropy adaptive reweighting for uncertainty-aware supervision allocation.

- Extensive experiments across multiple model families and vocabulary overlap ratios demonstrate that our method achieves state-of-the-art performance. Crucially, it exhibits superior robustness, consistently outperforming SFT in the evaluated low-overlap scenarios where competing methods fail.

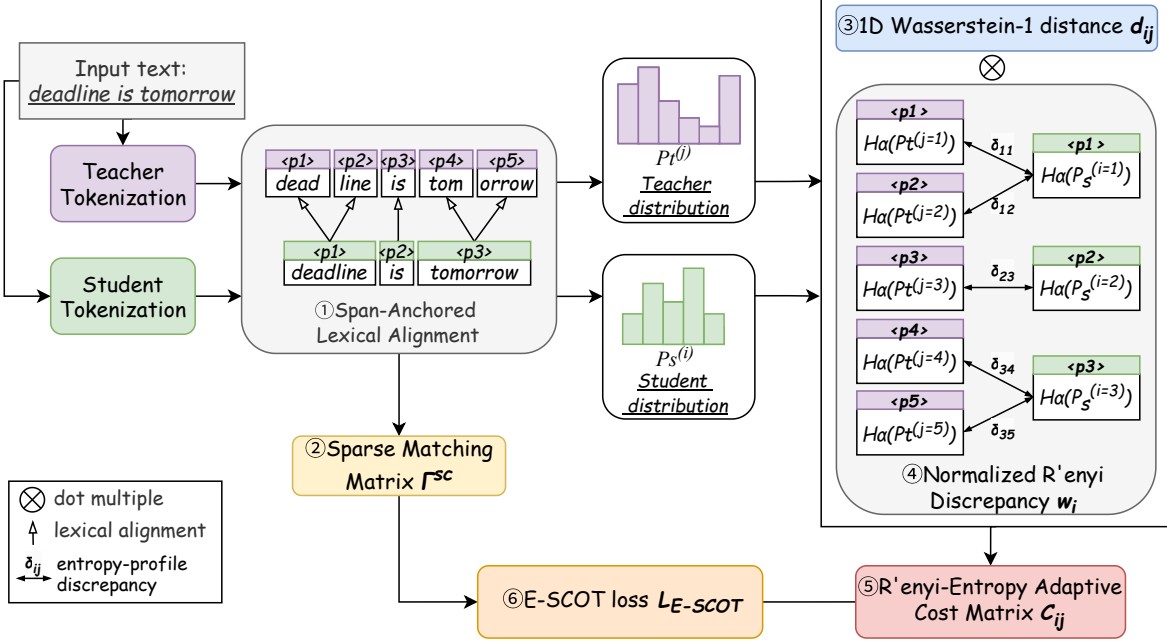

*Figure 3.* **Overview of the E-SCOT framework.** Given the same raw input text, we first establish a sparse coupling via **Span-Anchored Lexical Alignment** to resolve tokenizer mismatch. Subsequently, we compute a **Rényi-Entropy Adaptive Cost Matrix** to dynamically prioritize informative token pairs based on uncertainty divergence. This design effectively incorporates token-level and sequence-level information, ensuring robust knowledge transfer even under low vocabulary overlap.

## 2. Related Work

**Knowledge distillation for LLM.** Knowledge Distillation (KD) (Hinton et al., 2015) approaches are categorized into black-box and white-box paradigms. Black-box methods (Kim & Rush, 2016; Wang et al., 2023) utilize teacher-generated text for supervised fine-tuning but ignore fine-grained uncertainty. White-box methods, conversely, leverage internal probability distributions to enhance generation quality. To mitigate the mode-averaging inherent in standard forward KL, MiniLLM (Gu et al., 2024) minimizes reverse KL divergence, while GKD (Agarwal et al., 2024) addresses train-inference distribution shifts via on-policy distillation. Furthermore, DistiLLM (Ko et al., 2024) introduces adaptive objectives to improve gradient estimation and optimization efficiency. Despite these advancements, these methods fundamentally assume identical tokenizers between teacher and student, a restrictive constraint given the diversity of modern open-source LLMs.

**Cross-tokenizer KD for LLM.** Distilling knowledge across models with disparate vocabularies presents a significant challenge due to the lack of direct logit correspondence. Early attempts relied on heuristic vocabulary mappings or embedding alignment, which often result in semantic degradation. Recent state-of-the-art methods aim to establish probabilistic mappings. DSKD (Zhang et al., 2024b) introduces a dual-space framework that projects the student's hidden states into the teacher's space (and vice-versa) to compute divergence losses. MinED (Wan et al., 2024) uses dynamic time warping with an edit-distance-based cost to establish one-to-one token sequence alignment, and on the vocabulary level it augments exact-match mappings by searching for the most similar tokens according to edit distance to extend coverage. CDM (Chen et al., 2025) further advances the explicit-alignment paradigm by combining entropy-weighted dynamic time warping for sequence alignment with contextual dynamic Top-$K$ vocabulary mapping. These explicit alignment strategies can inject noise, particularly when the vocabulary overlap is low. This noise often outweighs the distillation benefits, causing the student to underperform the SFT baseline.

**Optimal Transport-based CTKD.** Optimal Transport (OT) offers a robust mathematical framework (Peyré & Cuturi, 2019) for comparing distributions without requiring point-to-point correspondence. Based on the OT theory, ULD (Boizard et al., 2025) proposes a Universal Logit Distillation loss, enabling distillation across arbitrary tokenizers without learnable projections. However, ULD reduces the OT problem to token-level alignment, without explicitly exploiting the local span-overlap structure induced by different tokenizations of the same raw text. This overlooks the sequential dependencies and structural context crucial for

language modeling. MultiLevelOT (Cui et al., 2025) extends ULD by jointly applying optimal transport at both token and sequence levels, employing multiple cost formulations (e.g., absolute and logarithmic difference costs) and approximating sequence-level Wasserstein distances via Sinkhorn iterations. While this strategy enhances robustness in certain settings, it retains a global, dense transport plan that lacks explicit structural constraints aligned with linguistic locality and monotonicity, requires careful tuning of multiple cost types and truncation thresholds, and still incurs substantial computation due to iterative Sinkhorn optimization.

## 3. Methods

To overcome the computational bottlenecks and structural looseness of global dense transport plans, we propose E-SCOT as an alternative solution. As illustrated in Figure 3, our framework operates through three cohesive components: (i) we adopt a **vocabulary-agnostic ground metric** to compare distribution shapes without shared tokens; (ii) we establish a **span-anchored alignment** via character overlaps to construct a sparse transport plan, avoiding expensive dense optimization; and (iii) we introduce a **Reńyi-entropy adaptive reweighting** to dynamically allocate distillation budgets to informative positions based on uncertainty discrepancy.

### 3.1. Optimal Transport Setup

Let $\mathbf{x}$ denote the same raw text. The teacher $f_T$ and student $f_S$ tokenize $\mathbf{x}$ into token sequences with vocabularies $V_T$ and $V_S$ ($|V_T| \neq |V_S|$). At teacher position $j$ and student position $i$, we compute temperature-scaled next-token distributions:

$$\mathbf{p}_T^{(j)} = \mathrm{softmax}(\mathbf{z}_T^{(j)}/\tau), \quad \mathbf{p}_S^{(i)} = \mathrm{softmax}(\mathbf{z}_S^{(i)}/\tau). \quad (1)$$

Let $\mathcal{I}$ and $\mathcal{J}$ be the supervised (non-padding) position sets for the student and teacher. We aggregate per-pair discrepancies between $\mathbf{p}_S^{(i)}$ and $\mathbf{p}_T^{(j)}$ using a nonnegative *matching matrix* $\Gamma \in \mathbb{R}_+^{|\mathcal{I}| \times |\mathcal{J}|}$ and a *cost matrix* $C \in \mathbb{R}_+^{|\mathcal{I}| \times |\mathcal{J}|}$:

$$\mathcal{L}_{\mathrm{tr}} = \langle \Gamma, C \rangle = \sum_{i \in \mathcal{I}} \sum_{j \in \mathcal{J}} \Gamma_{ij} C_{ij}. \quad (2)$$

Classical OT would solve for an optimal $\Gamma$ under marginal constraints. Here we *do not* optimize an outer transport plan; instead, we construct a sparse $\Gamma$ from a structural alignment prior and evaluate (2) under this fixed matching.

### 3.2. Vocabulary-Agnostic Ground Metric

To address vocabulary mismatch ($|V_S| \neq |V_T|$), we adopt the ULD ground metric (Boizard et al., 2025). The core insight is to compare *distribution shapes* rather than token identities: instead of aligning vocabulary indices, we treat

each next-token distribution as a *confidence profile* defined by its probability values.

Concretely, ULD measures the discrepancy between a student distribution and a teacher distribution by matching their *ranked* probabilities. This is equivalent to the **1D Wasserstein-1** distance on the sorted probability values and admits a closed-form expression (Boizard et al., 2025):

$$d_{ij} := W_1(\mathbf{p}_S^{(i)}, \mathbf{p}_T^{(j)})$$
$$= \frac{1}{\ell} \|\mathrm{sort}(\mathbf{p}_S^{(i)})^\uparrow - \mathrm{sort}(\mathbf{p}_T^{(j)})^\uparrow\|_1, \quad (3)$$

where $\ell = \max(|V_S|, |V_T|)$ and the shorter vector is zero-padded to match the length $\ell$. This metric effectively aligns the "sharpness" and "shape" of the predictive distributions without requiring a shared vocabulary. While the original ULD formulation applies this metric strictly time-step-wise (only computing $d_{ii}$), we employ it here as a general pairwise transport cost.

### 3.3. Span-Anchored Lexical Alignment

Since the teacher and student models process the same raw text, the character indices provide a natural, shared coordinate system for alignment. This construction assumes that both tokenizers expose reliable character-level offset mappings with respect to the same underlying raw string. Instead of solving a computationally expensive, dense transport problem to discover relationships, we leverage this structural prior to explicitly define *which* teacher tokens should supervise each student token.

**Construction of Support Set.** We define a sparse **support set** $\mathcal{K}$ of valid teacher–student token pairs using character-span overlap on the shared raw text. Pairs outside $\mathcal{K}$ are masked out, so a student token only receives supervision from teacher tokens covering the same text region.

Formally, let $\mathrm{Span}(s_i) = [\mathrm{st}_i^S, \mathrm{ed}_i^S)$ and $\mathrm{Span}(t_j) = [\mathrm{st}_j^T, \mathrm{ed}_j^T)$ denote the character intervals of the $i$-th student and $j$-th teacher tokens. We include a pair $(i, j)$ in $\mathcal{K}$ when two spans overlap:

$$\mathcal{K} = \{(i, j) \mid \mathrm{Span}(s_i) \cap \mathrm{Span}(t_j) \neq \emptyset\}. \quad (4)$$

For each student token $i$, we define its teacher neighborhood $\mathcal{N}(i) = \{j \mid (i, j) \in \mathcal{K}\}$. Crucially, since token spans are naturally sorted, we identify these overlaps via a **linear two-pointer sweep** (Algorithm 1) rather than a quadratic search. This reduces the support-set construction complexity to $O(|\mathcal{I}| + |\mathcal{J}|)$, making the alignment process highly efficient.

**Example.** To demonstrate how a support set is constructed in our algorithm, we give an illustrative example. Consider the raw text segment `"unbelievable"` (length

12). Suppose the teacher tokenizer splits it into three subtokens `["un", "bel", "ievable"]`, while the student splits it into two `["un", "believable"]`. Based on character offsets:

- Student token $s_1$ (`"un"`, range $[0, 2)$) aligns perfectly with teacher token $t_1$.

- Student token $s_2$ (`"believable"`, range $[2, 12)$) physically overlaps with both teacher tokens $t_2$ (`"bel"`) and $t_3$ (`"ievable"`).

Our method constructs the support set $\mathcal{K} = \{(1, 1), (2, 2), (2, 3)\}$. Crucially, this allows $s_2$ to aggregate supervision signals from multiple teacher tokens ($t_2$ and $t_3$) while maintaining strict local semantic consistency.

---

**Algorithm 1** Span-Anchored Lexical Alignment

---

1: **Input:** Student spans $\{[\mathrm{st}_i^S, \mathrm{ed}_i^S]\}_{i=1}^{|\mathcal{I}|}$, Teacher spans $\{[\mathrm{st}_j^T, \mathrm{ed}_j^T]\}_{j=1}^{|\mathcal{J}|}$
2: **Output:** Teacher neighborhoods $\{\mathcal{N}(i)\}$ for each student token $i$
3: Initialize pointer $j \leftarrow 1$
4: **for** each student token $i = 1$ **to** $|\mathcal{I}|$ **do**
5: $\quad \mathcal{N}(i) \leftarrow \emptyset$
6: $\quad$ // Advance teacher pointer to the first potential overlap
7: $\quad$ **while** $j \leq |\mathcal{J}|$ **and** $\mathrm{ed}_j^T \leq \mathrm{st}_i^S$ **do**
8: $\qquad j \leftarrow j + 1$
9: $\quad$ **end while**
10: $\quad$ // Collect all overlapping teacher tokens
11: $\quad k \leftarrow j$
12: $\quad$ **while** $k \leq |\mathcal{J}|$ **and** $\mathrm{st}_k^T < \mathrm{ed}_i^S$ **do**
13: $\qquad \mathcal{N}(i) \leftarrow \mathcal{N}(i) \cup \{k\}$
14: $\qquad k \leftarrow k + 1$
15: $\quad$ **end while**
16: **end for**
17: **return** $\{\mathcal{N}(i)\}_{i \in \mathcal{I}}$

---

**Defining the Matching Matrix $\Gamma$.** Once the support set $\mathcal{K}$ is established, we must assign transport weights. Rather than learning these weights (which is unstable and costly), we adopt a uniform prior: we assume that a student token $i$ should receive equal supervision from all teacher tokens in its neighborhood $\mathcal{N}(i)$. We construct the sparse matching matrix $\Gamma^{SC}$ as:

$$\Gamma_{ij}^{SC} = \begin{cases} 1/|\mathcal{N}(i)|, & \text{if } j \in \mathcal{N}(i) \text{ (valid overlap)}, \\ 0, & \text{otherwise.} \end{cases} \quad (5)$$

This formulation effectively averages the teacher's signals within the aligned span, providing a robust span-level supervision target for the student.

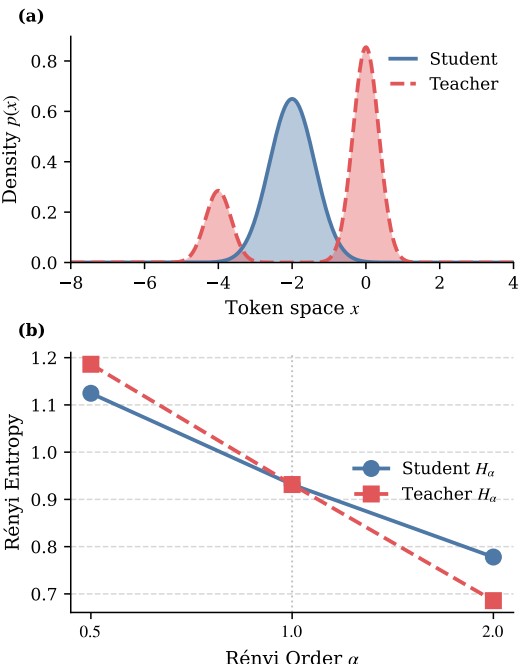

(a)

(b)

*Figure 4.* **A toy experiment: Reńyi Entropy Captures Structural Discrepancy.** (a) A unimodal Student and an asymmetric bimodal Teacher Gaussian distribution are constructed to share the exact same Shannon entropy ($H_1$). (b) While standard entropy metrics fail to distinguish them (intersection at $\alpha = 1$), the Rényi spectrum exposes a large gap at $\alpha \neq 1$, suggesting that $\alpha = 2$ and $\alpha = 0.5$ provide extra signals to align the geometric "shape" of two distributions.

### 3.4. Rényi-Entropy Adaptive Cost Matrix

Even within $\mathcal{N}(i)$, teacher and student may exhibit different uncertainty (confidence) profiles. We quantify this *uncertainty-profile gap* and treat it as an *informativeness* (hard-position) score: larger gaps indicate positions where the student deviates more from the teacher's confidence geometry, and empirically benefit more from distillation. Accordingly, we upweight such positions.

**Normalized Rényi discrepancy.** To ensure comparability across disparate vocabulary sizes ($|V_S| \neq |V_T|$), we employ the **normalized Rényi entropy**, which decouples the uncertainty measure from the dimension of the token space. For a probability distribution $\mathbf{p}$ over a vocabulary $V$ and order $\alpha \in \{0.5, 1, 2\}$, we define:

$$\hat{H}_\alpha(\mathbf{p}) = \frac{1}{\log |V|} \cdot \frac{1}{1 - \alpha} \log \left( \sum_{k=1}^{|V|} p_k^\alpha \right), \quad (6)$$

where the scaling factor $\frac{1}{\log |V|}$ maps the entropy to the unit interval $[0, 1]$. The case $\alpha = 1$ corresponds to the normalized Shannon entropy (limit as $\alpha \to 1$). Consequently, for a matched pair $(i, j) \in \mathcal{K}$, the scale-invariant entropy-profile

discrepancy is computed as:

$$\delta_{ij} = \sum_{\alpha} \left| \hat{H}_\alpha(\mathbf{p}_S^{(i)}) - \hat{H}_\alpha(\mathbf{p}_T^{(j)}) \right|, \quad q_i = \frac{1}{|\mathcal{N}(i)|} \sum_{j \in \mathcal{N}(i)} \delta_{ij}. \tag{7}$$

In language modeling, Shannon entropy can serve as a standard proxy for quantifying predictive uncertainty and estimating learning-difficulty of a token (Lin et al., 2024; Su et al., 2025; Wang et al., 2026). Normalized Rényi entropy can capture more overall uncertainty information of a distribution over token Shannon entropy, as shown in a toy example in Figure 4.

We compute a normalized token weight over matched positions:

$$w_i = \lambda + (1 - \lambda) \frac{|\mathcal{M}| \cdot q_i}{\sum_{k \in \mathcal{M}} q_k}, \quad \forall i \in \mathcal{M}, \tag{8}$$

where $|\mathcal{M}|$ denotes the total number of matched student tokens and ensures that the adaptive component is normalized to have a unit mean. $\lambda \in [0, 1]$ is a smoothing hyperparameter. This parameter acts as an interpolation coefficient that modulates the intensity of the adaptivity: $\lambda = 1$ recovers standard uniform weighting, while $\lambda \to 0$ yields fully adaptive weighting based on the entropy gap. We set $w_i = 1$ for $i \notin \mathcal{M}$. We stop gradients through $w_i$, using it purely as a loss reweighting coefficient. Intuitively, $w_i$ allocates more distillation budget to tokens with larger uncertainty-profile gaps. We obtain the Rényi-Entropy Adaptive Cost Matrix by $C_{ij} = w_i d_{ij}$. Thus, the E-SCOT distillation loss is:

$$\mathcal{L}_{\text{E-SCOT}} = \sum_{(i,j) \in \mathcal{K}} \Gamma_{ij}^{SC} C_{ij} \tag{9}$$

$$= \sum_{i \in \mathcal{M}} w_i \left( \frac{1}{|\mathcal{N}(i)|} \sum_{j \in \mathcal{N}(i)} d_{ij} \right). \tag{10}$$

### 3.5. Overall Training Loss

We apply E-SCOT as the distillation objective during SFT stage. In this stage, given an input sequence, the student model is trained with the standard language modeling cross-entropy loss on the supervised token-positions set $\mathcal{I}$

$$\mathcal{L}_{\text{CE}} = -\sum_{i \in \mathcal{I}} \log p_S(y_i \mid \mathbf{x}, y_{<i}), \tag{11}$$

where $y_i$ is the ground-truth next token at position $i$, and $p_S(\cdot \mid \mathbf{x}, y_{<i})$ is the student next-token distribution.

We combine the supervised loss with our distillation loss $\mathcal{L}_{\text{E-SCOT}}$:

$$\mathcal{L} = (1 - \rho) \mathcal{L}_{\text{CE}} + \rho \mathcal{L}_{\text{E-SCOT}}, \tag{12}$$

where knowledge distillation rate $\rho \in [0, 1]$ controls the strength of distillation.

## 4. Experiments

### 4.1. Experiment Setup

**Datasets and Metrics.** Following the widely-adopted setup (Gu et al., 2024; Ko et al., 2024), we use the Dolly-15k dataset (Ouyang et al., 2022) for training. We split the dataset into 14K training samples, 500 validation samples, and 500 test samples. We evaluate on five instruction-following benchmarks: Dolly, Self-Instruct (Wang et al., 2023), Vicuna-Evaluation (Chiang et al., 2023), Super-Natural Instructions (S-NI) (Wang et al., 2022), and Unnatural Instructions (UnNI) (Honovich et al., 2023). We use the Rouge-L (Lin, 2004) metric to evaluate the model-generated results and report the average scores of 5 generations for each prompt with different random seeds (10, 20, 30, 40, 50) for all test datasets. For the additional evaluation beyond the Dolly-15K setup, we train on a randomly sampled 20K subset of UltraChat-200K (Ding et al., 2023) and report results on four standard capability benchmarks: PiQA (Bisk et al., 2020), ARC-C (Clark et al., 2018), MMLU (Hendrycks et al., 2021), and AGI-CN (Zhong et al., 2024).

**Main Teacher–Student Model Pairs.** To validate the generalizability of our framework across varying degrees of vocabulary mismatch, we use a wide range of modern LLMs as teacher–student pairs: **Phi-3-mini-3.8B** (Abdin et al., 2024) → **Qwen2-0.5B** (Yang et al., 2024), **Phi-3-mini-3.8B** → **OPT-1.3B** (Zhang et al., 2022), **Llama-3-8B** (Grattafiori et al., 2024) → **Gemma-2-2B** (Team et al., 2024), and **Mistral-7B** (Jiang et al., 2023) → **TinyLlama-1.1B** (Zhang et al., 2024a). To demonstrate the performance degradation of existing CTKD methods compared with SFT, we also use Mistral-7B to distill Qwen2-0.5B, use Qwen2.5-7B to distill TinyLlama. The detailed vocabulary overlap ratios of the main model pairs are shown in Figure 1.

**Baselines.** In our experiments, the primary baseline is SFT applied to the student model. We benchmark E-SCOT against several advanced cross-tokenizer knowledge distillation methods: ULD (Boizard et al., 2025), Multi-LevelOT (Cui et al., 2025), DSKD (Zhang et al., 2024b), and MinED (Wan et al., 2024). We also compare with the black-box KD method SeqKD (Kim & Rush, 2016).

**Training Configurations.** To balance computational efficiency with model capacity, we adopt distinct optimization strategies based on model scale. We employ **Full Fine-Tuning (FFT)** for both teacher and student models in smaller pairs (e.g., Phi-3 → Qwen2-0.5B/OPT-1.3B). For larger pairs (e.g., Llama-3 → Gemma-2-2B, Mistral/Qwen2.5 → TinyLlama), we utilize LoRA for both models. Notably, for the Mistral-7B → Qwen2-0.5B pair, we adopt a hybrid setup where the teacher is fine-tuned via LoRA while the student undergoes FFT. For a fair comparison in the Dolly-15K experiments, we first fine-tune the

*Table 1.* Main results of comparing Ours and the baseline models. "Avg." means the average score of the instruction-following tasks. The **bold** text denotes the best performance in comparable cross-tokenizer distillation settings.

| Methods | Dolly | Self-Inst | Vicuna Eval | S-NI | UnNI | Avg. |
|---|---|---|---|---|---|---|
| \multicolumn{7}{c}{Phi-3-mini-3.8B → Qwen2-0.5B} | | | | | | |
| Teacher SFT | $30.89_{\pm 0.43}$ | $25.65_{\pm 1.29}$ | $23.94_{\pm 0.19}$ | $40.98_{\pm 0.20}$ | $40.28_{\pm 0.19}$ | 32.35 |
| SFT | $25.66_{\pm 0.11}$ | $17.13_{\pm 0.50}$ | $18.50_{\pm 0.29}$ | $31.95_{\pm 0.15}$ | $33.53_{\pm 0.09}$ | 25.35 |
| SeqKD | $25.43_{\pm 0.27}$ | $18.04_{\pm 0.21}$ | $18.52_{\pm 0.54}$ | $32.43_{\pm 0.27}$ | $34.15_{\pm 0.13}$ | 25.71 |
| ULD | $25.69_{\pm 0.27}$ | $18.53_{\pm 0.78}$ | $18.81_{\pm 0.35}$ | $32.78_{\pm 0.19}$ | $32.61_{\pm 0.10}$ | 25.68 |
| MultiLevelOT | $24.62_{\pm 0.28}$ | $17.03_{\pm 0.63}$ | $18.18_{\pm 0.33}$ | $30.80_{\pm 0.40}$ | $32.15_{\pm 0.08}$ | 24.56 |
| DSKD | $26.02_{\pm 0.28}$ | $16.24_{\pm 0.25}$ | $17.40_{\pm 0.49}$ | $29.99_{\pm 0.14}$ | $31.63_{\pm 0.19}$ | 24.26 |
| MinED | $25.06_{\pm 0.60}$ | $16.88_{\pm 0.35}$ | $16.81_{\pm 0.33}$ | $31.56_{\pm 0.20}$ | $31.39_{\pm 0.18}$ | 24.34 |
| Ours | $\mathbf{26.46}_{\pm 0.24}$ | $\mathbf{19.41}_{\pm 0.37}$ | $\mathbf{19.41}_{\pm 0.67}$ | $\mathbf{34.44}_{\pm 0.35}$ | $\mathbf{34.83}_{\pm 0.22}$ | **26.91** |
| \multicolumn{7}{c}{Llama-3-8B → Gemma-2-2B} | | | | | | |
| Teacher SFT | $31.25_{\pm 0.39}$ | $22.74_{\pm 0.89}$ | $19.79_{\pm 0.43}$ | $33.67_{\pm 0.34}$ | $39.10_{\pm 0.07}$ | 29.31 |
| SFT | $25.46_{\pm 0.57}$ | $18.65_{\pm 0.33}$ | $17.81_{\pm 0.24}$ | $33.28_{\pm 0.16}$ | $30.86_{\pm 0.23}$ | 25.21 |
| SeqKD | $26.70_{\pm 0.23}$ | $20.16_{\pm 0.98}$ | $18.50_{\pm 0.52}$ | $32.31_{\pm 0.12}$ | $31.71_{\pm 0.12}$ | 25.87 |
| ULD | $28.34_{\pm 0.58}$ | $20.84_{\pm 0.98}$ | $19.24_{\pm 0.20}$ | $34.97_{\pm 0.15}$ | $33.05_{\pm 0.15}$ | 27.29 |
| MultiLevelOT | $25.60_{\pm 0.46}$ | $19.22_{\pm 0.96}$ | $18.73_{\pm 0.30}$ | $32.67_{\pm 0.43}$ | $31.48_{\pm 0.12}$ | 25.58 |
| DSKD | $24.00_{\pm 0.71}$ | $17.33_{\pm 0.92}$ | $16.86_{\pm 0.34}$ | $26.79_{\pm 0.25}$ | $28.17_{\pm 0.12}$ | 22.63 |
| MinED | $27.57_{\pm 0.42}$ | $21.58_{\pm 0.97}$ | $\mathbf{19.35}_{\pm 0.60}$ | $34.51_{\pm 0.30}$ | $33.34_{\pm 0.10}$ | 27.27 |
| Ours | $\mathbf{28.76}_{\pm 0.52}$ | $\mathbf{22.02}_{\pm 0.43}$ | $19.18_{\pm 0.28}$ | $\mathbf{37.03}_{\pm 0.34}$ | $\mathbf{35.36}_{\pm 0.16}$ | **28.47** |
| \multicolumn{7}{c}{Phi-3-mini-3.8B → OPT-1.3B} | | | | | | |
| Teacher SFT | $30.89_{\pm 0.43}$ | $25.65_{\pm 1.29}$ | $23.94_{\pm 0.19}$ | $40.98_{\pm 0.20}$ | $40.28_{\pm 0.19}$ | 32.35 |
| SFT | $24.75_{\pm 0.32}$ | $14.33_{\pm 0.77}$ | $16.32_{\pm 0.56}$ | $26.46_{\pm 0.42}$ | $28.31_{\pm 0.16}$ | 22.03 |
| SeqKD | $24.81_{\pm 0.30}$ | $13.13_{\pm 0.42}$ | $15.67_{\pm 0.56}$ | $25.18_{\pm 0.66}$ | $28.44_{\pm 0.08}$ | 21.45 |
| ULD | $26.13_{\pm 0.41}$ | $14.05_{\pm 0.92}$ | $16.84_{\pm 0.46}$ | $27.70_{\pm 0.28}$ | $29.79_{\pm 0.12}$ | 22.90 |
| MultiLevelOT | $24.62_{\pm 0.32}$ | $13.46_{\pm 0.60}$ | $16.42_{\pm 0.29}$ | $25.27_{\pm 0.45}$ | $27.35_{\pm 0.11}$ | 21.42 |
| DSKD | $26.97_{\pm 0.38}$ | $\mathbf{14.78}_{\pm 0.52}$ | $18.12_{\pm 0.15}$ | $27.18_{\pm 0.20}$ | $29.64_{\pm 0.14}$ | 23.34 |
| MinED | $26.43_{\pm 0.28}$ | $14.71_{\pm 0.90}$ | $\mathbf{18.13}_{\pm 0.23}$ | $28.27_{\pm 0.33}$ | $30.60_{\pm 0.20}$ | 23.63 |
| Ours | $\mathbf{27.15}_{\pm 0.37}$ | $14.77_{\pm 0.63}$ | $17.19_{\pm 0.30}$ | $\mathbf{28.68}_{\pm 0.32}$ | $\mathbf{30.96}_{\pm 0.19}$ | **23.75** |
| \multicolumn{7}{c}{Mistral-7B → TinyLlama-1.1B} | | | | | | |
| Teacher SFT | $31.15_{\pm 0.24}$ | $25.10_{\pm 0.26}$ | $20.89_{\pm 0.37}$ | $38.22_{\pm 0.18}$ | $38.74_{\pm 0.14}$ | 30.82 |
| SFT | $23.12_{\pm 0.35}$ | $16.62_{\pm 0.49}$ | $16.61_{\pm 0.30}$ | $27.80_{\pm 0.43}$ | $27.48_{\pm 0.06}$ | 22.33 |
| SeqKD | $24.52_{\pm 0.33}$ | $16.63_{\pm 0.63}$ | $16.66_{\pm 0.33}$ | $27.84_{\pm 0.53}$ | $28.03_{\pm 0.15}$ | 22.74 |
| ULD | $25.37_{\pm 0.39}$ | $17.26_{\pm 0.34}$ | $17.70_{\pm 0.18}$ | $\mathbf{32.65}_{\pm 0.18}$ | $30.78_{\pm 0.07}$ | 24.75 |
| MultiLevelOT | $24.39_{\pm 0.56}$ | $17.77_{\pm 0.89}$ | $17.01_{\pm 0.13}$ | $28.61_{\pm 0.24}$ | $28.05_{\pm 0.15}$ | 23.17 |
| DSKD | $25.90_{\pm 0.54}$ | $18.68_{\pm 0.96}$ | $17.71_{\pm 0.55}$ | $32.30_{\pm 0.26}$ | $31.88_{\pm 0.12}$ | 25.29 |
| MinED | $25.74_{\pm 0.68}$ | $18.42_{\pm 0.43}$ | $\mathbf{17.81}_{\pm 0.62}$ | $31.51_{\pm 0.23}$ | $30.21_{\pm 0.08}$ | 24.74 |
| Ours | $\mathbf{26.76}_{\pm 0.30}$ | $\mathbf{18.74}_{\pm 0.27}$ | $17.54_{\pm 0.48}$ | $32.28_{\pm 0.29}$ | $\mathbf{32.16}_{\pm 0.25}$ | **25.50** |

teacher on the task data and then freeze it to supervise the student in all distillation runs, while keeping the student-side configuration identical between the SFT and CTKD runs. Detailed hyperparameters for the main Dolly-15K configurations are provided in Appendix A.

In particular, we set the distillation temperature $\tau$ to 2.0 and KD rate $\rho$ to 0.5 for all CTKD methods and set $\lambda$ to 0.25 for our method. For the Dolly-15K experiments, all methods are trained for 10 epochs, and the checkpoint with the highest Rouge-L score on the Dolly validation set is selected for evaluation. To ensure a fair comparison, we re-

implement all baselines using the official open-source code[1] and evaluate all methods under the same implementation and computational settings.

## 4.2. Main Results

**Low-Overlap negative transfer regime.** We demonstrate that existing CTKD methods do not always outperform even the SFT baseline in particular when the vocabulary overlap ratio is low. To make results comparable across different model families, we quantify the improvement of

---
[1] https://github.com/songmzhang/DSKD

each distillation method over the student SFT baseline using the normalized distillation gain:

$$\Delta_{sft} = \frac{R_{CTKD} - R_{sft}^S}{R^T - R_{sft}^S}, \quad (13)$$

where $R_{CTKD}$ denotes the performance of this distillation method, $R_{sft}^S$ denotes the performance of the SFT of student model, and $R^T$ denotes the performance of the teacher model. All $R$ scores are average Rouge-L scores over the five instruction-following tasks. A positive $\Delta_{sft}$ indicates that the CTKD method outperforms the SFT baseline, while a negative value indicates negative transfer. In this context ($\Delta_{sft} < 0$), cross-vocabulary distillation yields negative gains while incurring higher computational overhead. As shown in Figure 2, MinED and DSKD exhibit negative transfer when the vocabulary overlap between the teacher and student models is low. While ULD remains relatively robust in such settings, it generally lags behind alignment-based methods in scenarios with higher overlap.

**Comparison with baselines.** As shown in Table 1, our proposed method consistently outperforms baselines across all four diverse teacher–student pairs, achieving the highest average scores in every setting. Specifically, our method achieves a **4.67%** improvement for Qwen2-0.5B and a **4.32%** improvement for Gemma-2-2B in average scores compared to the best-performing baseline. Crucially, our approach demonstrates superior robustness compared to prior cross-tokenizer distillation methods: (1) It consistently exceeds OT-based baselines (ULD and MultiLevelOT), confirming that our *span-constrained* matching effectively captures local structural dependencies that token-level OT misses; (2) It outperforms explicit alignment methods (DSKD and MinED), avoiding the performance degradation often caused by heuristic misalignment errors in low-overlap regimes. Furthermore, our method requires neither auxiliary storage for vocabulary mapping nor additional trainable parameters. On the Mistral $\rightarrow$ TinyLlama pair, which exhibits the highest vocabulary overlap, our method maintains stable improvements, while complex baselines like MultiLevelOT struggle to surpass the simpler ULD.

**Evaluation beyond Dolly-15K.** To further examine whether the advantage of E-SCOT extends beyond the original Dolly-15K setup, we conduct an additional experiment on a larger and more recent teacher–student pair, **Phi-4-14B $\rightarrow$ Qwen3-4B**. Specifically, we train on a randomly sampled 20K subset of UltraChat-200K with a maximum sequence length of 1024 for 3 epochs; the tokenizer overlap ratio for this pair is 64.22%. We evaluate the resulting models on four capability benchmarks, including PiQA, ARC-C, MMLU, and AGI-CN. As shown in Table 2, E-SCOT outperforms both SFT and DSKD in average score. Notably, DSKD remains below SFT on average, indicating that nega-

*Table 2.* **Evaluation beyond Dolly-15K.** Results on Phi-4-14B $\rightarrow$ Qwen3-4B trained with a 20K subset of UltraChat-200K. Bold indicates the best result among student-side methods. Bold indicates the best result among fine-tuned or distilled student methods.

| Method | PiQA | ARC-C | MMLU | AGI-CN | Avg. |
|--------|------|-------|------|--------|------|
| Teacher | 0.8128 | 0.5606 | 0.7691 | 0.5750 | 0.6794 |
| Base | 0.7514 | 0.5410 | 0.6832 | 0.5482 | 0.6310 |
| SFT | 0.7617 | 0.5546 | 0.6763 | 0.6067 | 0.6498 |
| DSKD | 0.7622 | 0.5427 | 0.6736 | 0.5935 | 0.6430 |
| E-SCOT | **0.7769** | **0.5606** | **0.6800** | **0.6227** | **0.6601** |

tive transfer can still occur beyond the original Dolly-15K setting. These results provide additional evidence that the advantage of E-SCOT is not limited to the original Dolly-15K training and Rouge-L evaluation setup.

*Table 3.* **Ablation Study.** We report the average performance across 5 downstream tasks. **Align.**: Alignment Support; **Idx-to-Idx**: Index-to-Index; **Span-Anch.**: Span-Anchored.

| Method Variant | Align. | Reweight. | Avg. Perf. Score | $\Delta$ |
|----------------|--------|-----------|------------------|----------|
| **ULD (baseline)** | Idx-to-Idx | Uniform | **25.68** | – |
| **E-SCOT (Ours)** | **Span-Anch.** | **Rényi** | **26.91** | +1.23 |
| *Impact of Alignment* | | | | |
| w/o Span Constraint | Idx-to-Idx | Rényi | 26.51 | +0.97 |
| w/ Max-Overlap | Span (Max) | Rényi | 26.88 | +1.20 |
| *Impact of Reweighting* | | | | |
| w/o Adaptivity ($\lambda$=1) | Span-Anch. | Uniform | 26.00 | +0.32 |
| w/ Shannon ($\alpha$=1) | Span-Anch. | Shannon | 26.18 | +0.50 |

### 4.3. Ablation Study

We conduct experimental analyses to verify the effectiveness of each component of our method through Phi3-mini-3.8B distilling Qwen2-0.5B in instruction-following datasets.

We adopt ULD as the baseline, which employs the same vocabulary-agnostic Wasserstein metric but relies on a rigid index-to-index alignment with uniform weighting. As shown in Table 3, both span-anchored lexical alignment (**Span-Anchored**) and Rényi-entropy adaptive reweighting consistently improve upon ULD, and the gain from Rényi-entropy adaptive reweighting is more pronounced. Combining all components achieves the best overall performance across evaluation sets, suggesting that span-based locality and Rényi-entropy adaptive reweighting are complementary. We further compare normalized Shannon entropy with normalized Rényi entropy for constructing the adaptive cost matrix. The Rényi-based variant achieves better performance, suggesting that multi-order entropies capture distributional shape differences beyond $\hat{H}_1$. Finally, replacing the uniform **Span-Anchored** aggregation with the max-overlap variant (Span (Max)) yields comparable gains, suggesting that the main benefit comes from enforcing span-

based locality rather than from the specific aggregation rule within each local neighborhood.

**Impact of parameter** $\lambda$**.** We conduct a sensitivity analysis to investigate the impact of the hyperparameter $\lambda$ in Rényi-entropy adaptive cost matrix, as shown in Figure 5. The red curve depicts the macro-average score aggregated across all five benchmarks (Dolly, S-NI, Vicuna-Eval, Self-Inst, and UnNI). We observe that the optimal overall performance is achieved at $\lambda = 0.25$, achieving a balance between stability ($\lambda \rightarrow 1$) and informativeness ($\lambda \rightarrow 0$).

Additional experimental results, including efficiency analysis, LLM-as-judge evaluation, same-vocabulary KD experiments, and high-overlap $\lambda$ sensitivity studies, are provided in Appendix B.

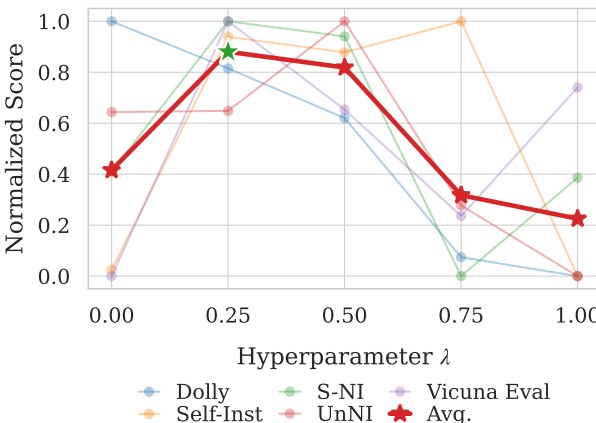

*Figure 5.* Impact of varying $\lambda$ on normalized Rouge-L scores: results for five datasets and the dataset-level average.

## 5. Conclusion

In this work, we identify the **Low-Overlap negative transfer regime** in cross-tokenizer distillation, revealing that explicit alignment strategies often fail to outperform standard SFT. To address this, we propose **E-SCOT**, a robust framework that replaces expensive dense transport with a deterministic span-anchored lexical alignment and dynamically concentrates the distillation budget via Rényi-entropy adaptive reweighting. By enforcing local semantic consistency and prioritizing positions with significant uncertainty-profile gaps, E-SCOT effectively mitigates tokenizer-induced misalignment. Extensive experiments demonstrate that E-SCOT establishes a new state-of-the-art, consistently outperforming SFT in the evaluated low-overlap scenarios. Crucially, our approach achieves this with linear-time span-support construction and without requiring auxiliary parameters or additional storage overhead, suggesting a promising direction for efficient knowledge transfer in an increasingly heterogeneous LLM landscape.

## Impact Statement

This paper presents work whose goal is to advance efficient and robust knowledge distillation for large language models with heterogeneous tokenizers. By improving cross-tokenizer distillation, E-SCOT may facilitate capability transfer across different model families and reduce the cost of adapting smaller student models, especially when teacher and student tokenizers differ substantially. This may make capable language technologies more accessible in resource-constrained deployment settings. As with other distillation methods, the resulting student models may inherit undesirable behaviors from the teacher model or training data, and should therefore be evaluated carefully before deployment.

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

# A. Experimental Details.

## A.1. Training Configurations

We detail the hyperparameters in Table 4. The **Teacher** columns refer to the configuration used to fine-tune the teacher models (which are subsequently frozen). Crucially, to ensure a fair comparison, the **Student** configurations (including learning rate, batch size, and LoRA rank) are kept consistent across both the SFT baselines and other distillation experiments.

*Table 4.* **Detailed Training Hyperparameters.** Smaller model pairs use **Full Fine-Tuning**, while larger pairs use **LoRA**. Note that the student hyperparameters apply to both the SFT baseline training and the distillation process.

| Hyperparameters | Full Fine-Tuning | | LoRA Fine-Tuning | |
|---|---|---|---|---|
| | **Teacher** *(SFT Only)* | **Student** *(SFT & Distill)* | **Teacher** *(SFT Only)* | **Student** *(SFT & Distill)* |
| *Model Pairs (T → S)* | Phi-3-mini-3.8B → Qwen2-0.5B / OPT-1.3B | | Llama-3-8B / Mistral-7B → Gemma-2-2B / TinyLlama | |
| *Optimization* | | | | |
| Global Batch Size | 32 | 32 | 32 | 32 |
| Learning Rate | 2e-5 | 2e-5 | 1e-3 | 1e-3 |
| LR Scheduler | Cosine | Cosine | Cosine | Cosine |
| Max Seq. Length | 512 | 512 | 512 | 512 |
| Epochs | 10 | 10 | 10 | 10 |
| *LoRA Config* | | | | |
| LoRA Rank ($r$) | – | – | 256 | 256 |
| LoRA Alpha ($\alpha$) | – | – | 8 | 8 |
| LoRA Dropout | – | – | 0.1 | 0.1 |

## A.2. Hyperparameter Selection.

A critical hyperparameter in our E-SCOT framework is the smoothing coefficient $\lambda \in [0, 1]$ (Eq. 8), which controls the intensity of the entropy-adaptive reweighting.

- $\lambda \to 1$: The cost matrix reverts to a standard uniform weighting.

- $\lambda \to 0$: The cost becomes fully sensitive to the Rényi entropy discrepancy.

We conduct a sensitivity analysis on the Dolly validation set to determine the optimal $\lambda$. As illustrated in Figure 6, the student performance (Rouge-L) initially improves as adaptivity increases, peaking at $\lambda = 0.25$, before degrading when the reweighting becomes too aggressive ($\lambda < 0.1$). Consequently, we fix $\lambda = 0.25$ for all main experiments.

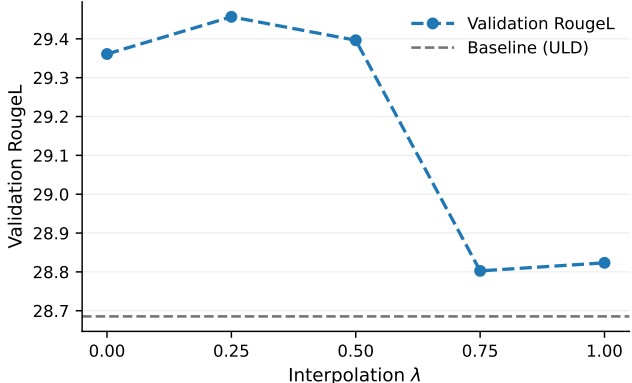

*Figure 6.* **Impact of the smoothing parameter $\lambda$ on student performance.** We report the Rouge-L score on the Dolly validation set. The curve demonstrates that a moderate degree of adaptivity ($\lambda = 0.25$) yields the best trade-off between stability and informativeness, outperforming both the fully uniform baseline ($\lambda = 1.0$) and the highly aggressive adaptive setting ($\lambda \to 0$).

# B. Additional Experiments

## B.1. Training Efficiency

To clarify the practical cost of E-SCOT, we compare peak allocated GPU memory, wall-clock step time, and training throughput under the same Llama-3-8B $\rightarrow$ Gemma-2-2B setting. As shown in Table 5, E-SCOT is more expensive than plain SFT, as expected for white-box CTKD, but remains comparable to existing CTKD baselines. In particular, E-SCOT uses less peak memory than ULD and MultiLevelOT, and achieves comparable throughput to other CTKD methods.

*Table 5.* **Training efficiency comparison.** Peak memory, step time, and throughput are measured under the same Llama-3-8B $\rightarrow$ Gemma-2-2B setting. Step time is measured as the maximum wall-clock time across ranks.

| Method | Peak Mem. (GB) | Step Time (s) | Throughput (tok/s) |
|---|---|---|---|
| SFT | 11.39 | 0.94 | 2570.99 |
| DSKD | 42.01 | 1.35 | 1784.03 |
| MultiLevelOT | 55.21 | 2.12 | 1156.08 |
| ULD | 42.99 | 2.57 | 1079.15 |
| E-SCOT | 41.57 | 2.31 | 1041.82 |
| MinED | 39.08 | 2.64 | 903.64 |

## B.2. LLM-as-Judge Evaluation

Since Rouge-L primarily measures surface-form overlap, we further conduct an LLM-as-Judge evaluation on Vicuna-Eval for the Phi-3-mini-3.8B $\rightarrow$ Qwen2-0.5B setting. We use Qwen2.5-Plus as the judge model and compare E-SCOT against SFT and representative CTKD baselines. As shown in Table 6, E-SCOT obtains higher win rates than all compared methods, providing additional evidence that its advantage is not limited to Rouge-L evaluation.

*Table 6.* **LLM-as-Judge evaluation** on Vicuna-Eval for Phi-3-mini-3.8B $\rightarrow$ Qwen2-0.5B. Win rate is computed by counting each tie as half a win.

| Comparison | Win Rate | Win/Tie/Loss |
|---|---|---|
| E-SCOT vs. SFT | 64.38% | 49 / 5 / 26 |
| E-SCOT vs. MinED | 66.25% | 48 / 10 / 22 |
| E-SCOT vs. ULD | 58.13% | 41 / 11 / 28 |
| E-SCOT vs. DSKD | 69.38% | 51 / 9 / 20 |

## B.3. Same-vocabulary KD with Rényi Reweighting

To examine whether Rényi-entropy adaptive reweighting can also be beneficial beyond CTKD, we additionally test it in a same-vocabulary KD setting, Llama2-7B $\rightarrow$ TinyLlama-1.1B. As shown in Table 7, adding Rényi reweighting improves over the skewed reverse KL baseline on all three benchmarks. This suggests that Rényi reweighting can serve as a useful token-level weighting strategy, while E-SCOT further combines it with span-local support for cross-tokenizer supervision.

*Table 7.* **Same-vocabulary KD with Rényi reweighting.** Results on Llama2-7B $\rightarrow$ TinyLlama-1.1B.

| Method | SuperNI | UnNI | Vicuna |
|---|---|---|---|
| Skewed Reverse KL | 31.19 | 30.43 | 17.01 |
| + Rényi Reweighting | **32.15** | **31.65** | **17.06** |

## B.4. Additional Sensitivity to $\lambda$

We further evaluate the sensitivity of $\lambda$ on the high-overlap Mistral-7B $\rightarrow$ TinyLlama setting, whose vocabulary overlap ratio is 75.6%. As shown in Table 8, smaller values of $\lambda$ generally perform better than larger values, while the best value can vary across evaluation sets. This suggests that overlap-aware tuning of $\lambda$ may be a useful future direction, although the default $\lambda = 0.25$ already performs robustly across the main experiments.

*Table 8.* **Additional sensitivity to** $\lambda$ on the high-overlap Mistral-7B $\rightarrow$ TinyLlama setting.

| $\lambda$ | Vicuna-Eval | SuperNI | UnNI |
|---|---|---|---|
| 0.1 | **17.72** | 31.93 | **32.39** |
| 0.25 | 17.54 | **32.28** | 32.16 |
| 0.5 | 16.97 | 30.76 | 30.87 |
| 0.75 | 17.07 | 29.64 | 30.43 |

### B.5. Details of the Toy Experiment: Rényi Entropy Captures Structural Discrepancy

We construct a pair of distributions with identical Shannon entropy but different structural properties to illustrate that the (normalized) Rényi entropy spectrum captures information beyond token entropy.

**Token space.** All distributions are defined on a shared one-dimensional token space $x \in [-8, 4]$, discretized uniformly. A constant shift $\text{SHIFT} = -2.0$ is applied to both distributions, which does not affect entropy comparisons.

**Teacher distribution.** The teacher is defined as an asymmetric bimodal Gaussian mixture:

$$p_T(x) = w\,\mathcal{N}(x; -s + \text{SHIFT},\ \sigma_T) + (1 - w)\,\mathcal{N}(x; +s + \text{SHIFT},\ \sigma_T),$$

with parameters

$$\sigma_T = 0.35, \quad s = 2.0, \quad w = 0.25.$$

This results in a sharp, asymmetric bimodal distribution.

**Student distribution.** The student is restricted to a unimodal Gaussian family:

$$p_S(x; \sigma) = \mathcal{N}(x; 0 + \text{SHIFT},\ \sigma),$$

where $\sigma$ is the only free parameter.

**Entropy matching.** We choose $\sigma$ by minimizing the squared difference in Shannon entropy:

$$\sigma^\star = \arg \min_{\sigma \in [0.5, 6.0]} \big(H_1(p_S) - H_1(p_T)\big)^2.$$

This ensures $H_1(p_S) \approx H_1(p_T)$ by construction.

**Rényi entropy evaluation.** We then compute Rényi entropies at orders

$$\alpha \in \{0.5,\ 1.0,\ 2.0\}.$$

Although the two distributions have matched $H_1$, they exhibit clearly different $H_{0.5}$ and $H_2$ values due to their distinct structural shapes (unimodal vs. asymmetric bimodal), as shown in Figure 4.

