# OpenReview forum: "Entropy-aware Span-Constrained Optimal Transport for Robust Cross-Tokenizer Knowledge Distillation"
_ICML.cc/2026/Conference — ICML 2026 regular_

### Official Review · Reviewer_rCoh · 2026-03-08

**Soundness:** 3
**Presentation:** 4
**Significance:** 3
**Originality:** 2
**Overall Recommendation:** 5
**Confidence:** 4

**Summary:**

This paper addresses Cross-Tokenizer Knowledge Distillation (CTKD) — distilling knowledge from a teacher LLM to a student LLM that uses a different tokenizer and vocabulary. Standard KD relies on comparing logit distributions over a shared vocabulary, which breaks when tokenizers differ. Two alignment problems must be solved: (1) sequence alignment — ensuring the teacher provides supervision at the correct temporal step, and (2) vocabulary alignment — mapping the teacher's probability distribution to the student's vocabulary space. The authors first identify the "Low-Overlap negative transfer regime": existing CTKD methods (MinED, DSKD, ULD) often underperform simple SFT when vocabulary overlap is low, because alignment noise overwhelms the distillation signal.

To address this, the paper proposes E-SCOT (Entropy-aware Span-Constrained Optimal Transport). The method maps teacher tokens to student tokens via character-span overlap on the shared raw text, building a sparse matching. It then uses Rényi entropy at multiple orders (α ∈ {0.5, 1, 2}) and the 1D Wasserstein-1 distance on sorted probability values (adopted from ULD) to distill the *shape* of the teacher's confidence profile — effectively transferring the teacher's uncertainty level into the student. Renyi entropy is used because it captures the shape of the distribtion better than a Shannon entropy. The entropy gap between teacher and student at each position serves as a weight: positions with larger uncertainty-profile gaps receive more distillation budget, while prior methods weight all positions uniformly. The E-SCOT loss is:

$$L_{\text{E-SCOT}} = \sum_{i \in M} w_i \left( \frac{1}{|N(i)|} \sum_{j \in N(i)} d_{ij} \right)$$

where $d_{ij}$ is the Wasserstein-1 distance between sorted teacher and student distributions, $N(i)$ is the set of teacher tokens overlapping student token $i$, and $w_i$ is the Rényi-entropy-based weight. This loss is then combined with the usual SFT cross-entropy loss: $L = (1-\rho) L_{CE} + \rho L_{\text{E-SCOT}}$.

The method combines character-span alignment with multi-order entropy for position importance. While both ideas have been used separately in prior work, this paper combines them for CTKD.

The authors show that E-SCOT is superior to alternatives on the Dolly-15k dataset, measuring RougeL across five instruction-following benchmarks and reporting average scores of 5 generations per prompt with different random seeds.

**Compliance With Llm Reviewing Policy:**

Affirmed.

**Ethical Review Concerns:**

The PDF contains unusual text on pages 2 and 12 that  can be found by searching the PDF for the string "The authors explore an important" I am not certain whether this is an intentional prompt injection targeting LLM-assisted reviews or simply a formatting/copy-paste artifact.
Update: I suspect it was just automatically added by the review system.

**Final Justification:**

After considering both the paper and the rebuttal, my final recommendation is Accept. The paper addresses a practically important problem in cross-tokenizer knowledge distillation, and I find the method technically sound and empirically effective within the reported setup. The paper is also generally clear, and the ablations help support the claimed contributions.

My main initial concerns were about evaluation breadth, experimental scale, efficiency, and the strength of some empirical claims. The rebuttal addressed these concerns meaningfully by adding a larger-scale experiment, broader benchmark results beyond instruction-following Rouge-L, and a runtime/memory comparison. These additions increased my confidence in the significance of the work, and I also appreciate that the authors clarified the framing of the OT component and moderated claims for small-margin gains.

Some weaknesses remain. The originality is still moderate, since the contribution is mainly a useful combination of existing ideas rather than a major conceptual advance, and I still think a direct comparison to more recent baselines would have strengthened the paper further. Nonetheless, the rebuttal changed my assessment: while I previously viewed the empirical case as too narrow, I now think the paper provides a solid and useful contribution to this subarea, and I am comfortable supporting acceptance.

**Key Questions For Authors:**

1. Could you provide significance test results for cases where the improvement over the best baseline is small?
2. Have you analyzed why "Span-Anchored" (uniform averaging) and "Span (Max)" (single best-overlap teacher token) perform almost identically in the ablation (+1.23 vs. +1.20)? Do you think there is a smarter way to combine tokens within a span?
3. It would be helpful to see a broader evaluation on additional benchmarks and with longer context lengths to understand how the method compares against the main competitors more broadly.
4. Could you add a training time comparison between the main methods?

**Limitations:**

yes

**Strengths And Weaknesses:**

### Soundness

**Strengths:**

- The method is technically sound. Results are supported with empirical validation against the main published competitive methods across five teacher–student pairs with vocabulary overlap ratios ranging from 15% to 76%. This systematic variation clearly demonstrates the low-overlap negative transfer problem and motivates the proposed solution.
- The ablation study cleanly isolates the contributions of each component: span-anchored alignment and Rényi-entropy reweighting are shown to be complementary, and the comparison between Rényi and Shannon entropy for reweighting provides useful insight into why multi-order entropy helps.

**Weaknesses:**

- The evaluation relies on a single, relatively small training dataset (Dolly-15k, 14K samples) and a single metric (RougeL on instruction-following tasks). There is no evaluation on standard LLM capability benchmarks (e.g., MMLU, ARC, GSM8K, HumanEval), making it hard to judge whether E-SCOT broadly preserves the teacher's knowledge or only improves on this specific setup. Competitors like CDM and ALM evaluate on math and code in addition to instruction-following.
- The models used are relatively small (up to 8B teacher, up to 2B student), and the max sequence length is only 512 tokens. It would be valuable to see how the method scales to larger models and longer contexts, where practical distillation increasingly matters and where the span alignment advantage over index-based matching should grow (positional drift accumulates with sequence length).
- The strategy for weighting within token neighborhoods is not sufficiently analyzed. The paper uses uniform weighting (Eq. 5), but alternatives — such as weighting proportionally to the number of overlapping characters — are not explored. Moreover, "Span-Anchored" (uniform averaging) and "Span (Max)" (single best-overlap teacher token) perform almost identically in the ablation (+1.23 vs. +1.20), suggesting that the specific weighting scheme matters little. This raises an unanswered question: is it the locality constraint that drives the gains, rather than how supervision is aggregated within the neighborhood?

### Presentation

**Strengths:**

- The paper is generally well-organized and readable. Figures and tables effectively convey the method, results, and ablations. In particular, Figure 3 gives a clear end-to-end overview, and the "unbelievable" example makes the span alignment intuitive.

### Significance

**Strengths:**

- Cross-tokenizer distillation is a practically important problem given the diversity of open-source LLM families (Llama, Qwen, Gemma, Phi, Mistral, etc.). A method that reliably avoids negative transfer in low-overlap regimes has clear utility.
- The paper provides empirical validation that combining text-anchored token alignment with distribution shape distillation (via Wasserstein distance) meaningfully improves knowledge transfer quality, offering a useful signal for future work in this area.
- Implementation details and hyperparameter choices are well described, aiding reproducibility.

**Weaknesses:**

- The improvements over the best baseline, while consistent, are often modest and may not be statistically significant. For example, Phi-3 → OPT-1.3B shows only +0.12 average RougeL over MinED (23.75 vs. 23.63), and Mistral → TinyLlama shows +0.21 over DSKD (25.50 vs. 25.29). Given the reported variance, these gaps are within noise range for some pairs.
- The experimental scale is limited: all models are ≤8B parameters with a 512-token context window. It is unclear whether the gains hold or increase in more realistic settings with larger models and longer sequences, which is where cross-tokenizer distillation is most needed in practice.
- No wall-clock time or computational cost comparisons are provided. The paper claims O(|I|+|J|) alignment complexity, but without reporting actual training time savings over, e.g., MultiLevelOT's iterative Sinkhorn optimization, the efficiency advantage remains theoretical.

### Originality

**Strengths:**

- While token-span-alignment methods and the vocabulary-agnostic Wasserstein metric exist independently in prior work, their combination — span-based alignment with Rényi-entropy-weighted distribution shape matching — is novel in the CTKD context to the best of my knowledge.

**Weaknesses:**

- One of the three core components (the 1D Wasserstein-1 ground metric on sorted probabilities) is entirely borrowed from ULD (Boizard et al., 2024), so the actual novelty rests on the other two components.

---

> ### Author Rebuttal · Authors · 2026-03-31
>
> Thank you for the thorough and balanced review. We are glad that you found the paper technically sound, clearly written, and practically relevant. Your concerns focus mainly on evaluation breadth, experimental scale, significance of gains, and the interpretation of the span-locality component; we address them below.
>
> 1-2. **Broader evaluation and newer/larger models.**
> We agree that the original submission was limited by a relatively small training dataset (Dolly-15K), a RougeL-based instruction-following evaluation, and modest model/context scales. To address this, we added a new experiment in a larger-model, longer-context setting: Phi-4-14B -> Qwen3-4B, trained on an UltraChat-200K subset with max sequence length 1024 for 3 epochs (tokenizer overlap ratio: 64.22%). We also broadened evaluation beyond instruction-following by testing PiQA, ARC-C, MMLU, and AGI-CN.
>
> | Phi4-14B -> Qwen3-4B | PiQA | ARC-C | MMLU | AGI-CN | Avg |
> |---|---:|---:|---:|---:|---:|
> | teacher | 0.8128 | 0.5606 | 0.7691 | 0.5750 | 0.6794 |
> | Base | 0.7514 | 0.5410 | 0.6832 | 0.5482 | 0.6310 |
> | SFT | 0.7617 | 0.5546 | 0.6763 | 0.6067 | 0.6498 |
> | DSKD | 0.7622 | 0.5427 | 0.6736 | 0.5935 | 0.6430 |
> | E-SCOT | 0.7769 | 0.5606 | 0.6800 | 0.6227 | 0.6601 |
>
> In this setting, E-SCOT outperforms both SFT and DSKD. DSKD remains below SFT on average (0.6430 vs. 0.6498), indicating that negative transfer can still arise beyond Dolly-15K. We believe this provides meaningful new evidence for broader benchmarks and newer/larger model settings.
>
> 3. **Weighting inside token neighborhoods.**
> We agree that the closeness between "Span-Anchored" and "Span (Max)" suggests that the main gain may come from enforcing locality itself rather than from the exact aggregation rule inside the neighborhood. We will revise the discussion accordingly. Our current evidence supports the view that span-constrained local support is the key ingredient, while the neighborhood-internal aggregation rule is a second-order choice. We also agree that more principled alternatives, such as overlap-proportional weighting, are worth investigating. Although we do not yet have a full additional study of these variants within the rebuttal timeline, we will make this limitation explicit and avoid claiming that uniform averaging is uniquely optimal.
>
> 4. **Small gains and statistical significance.**
> We appreciate this suggestion. Table 1 already reports mean +/- standard deviation over five test runs/generations, so the submission does include an explicit measure of variability. At the same time, we agree that for small-margin cases such as Phi-3 -> OPT-1.3B (+0.12) and Mistral -> TinyLlama (+0.21), mean +/- std alone does not establish strong statistical significance, and these cases should not be over-interpreted. We will therefore revise the wording to avoid overly strong claims for such modest gains.
>
> We also note that the strongest competing baseline is not the same across settings: different CTKD baselines perform best on different teacher-student pairs, whereas E-SCOT consistently achieves the best average performance across all four settings in Table 1. We believe this cross-setting consistency is the more important empirical pattern, and it is further supported by the additional Phi-4-14B -> Qwen3-4B experiment.
>
> 5. **Training cost.**
> We thank the reviewer for pointing out that the efficiency claim should be supported by actual runtime measurements rather than asymptotic complexity alone. To address this, we added a direct cost comparison under the same hardware/training configuration for the Llama3-8B -> Gemma2-2B setting.
>
> | Method | Peak Mem Alloc (GB) | Step Time (s, max across ranks) | Throughput (tok/s) |
> |---|---:|---:|---:|
> | SFT | 11.39 | 0.94 | 2570.99 |
> | DSKD | 42.01 | 1.35 | 1784.03 |
> | MultiLevelOT | 55.21 | 2.12 | 1156.08 |
> | ULD | 42.99 | 2.57 | 1079.15 |
> | E-SCOT | 41.57 | 2.31 | 1041.82 |
> | MinED | 39.08 | 2.64 | 903.64 |
>
> We also clarify that the 1D Wasserstein-1 ground metric is adopted from ULD and is not claimed as a novel contribution here. Our novelty lies in the span-anchored support construction and the Renyi-based reweighting. In this experiment, E-SCOT shows practical runtime cost comparable to strong CTKD baselines, while using substantially less peak memory than ULD and especially MultiLevelOT, and slightly less than DSKD. Compared with MultiLevelOT, E-SCOT reduces peak allocated memory from 55.21 GB to 41.57 GB while keeping similar step time and throughput. Compared with ULD and MinED, E-SCOT remains in a similar runtime regime. We will revise the wording so that the linear-time claim is used specifically for support-set construction rather than the entire end-to-end training pipeline.
>
> Thanks again for the review and suggestions. We believe the added large-scale, broader-benchmark, and efficiency results address the key remaining gaps.

---

> > ### Author Rebuttal · Reviewer_rCoh · 2026-04-04
> >
> > After the rebuttal:
> > Originality: the rebuttal does not improve the novelty much; if anything, it narrows the claim to "span-anchored support + Renyi reweighting," while the Wasserstein part is explicitly inherited. Still, it is a valuable contribution, as it strengthens the empirical results.
> >
> > Soundness has improved. The new larger-scale Phi-4 -> Qwen3 experiment and the broader benchmarks help verify the method's quality, and the efficiency table helps clarify the additional cost and shows that it is comparable to competing methods.
> >
> > Overall, I'm more confident about the paper now, and will improve my score to Accept.
> >
> > I still have one follow-up question. CDM (Chen et al., ACL Findings 2025) appears to be a relevant recent CTKD baseline with overlapping model pairs and instruction-following benchmarks. Could you clarify why it was not included in the experimental comparison, and whether you expect it to be competitive with E-SCOT under your exact setup? In addition, for the overlapping experiments, the absolute results seem noticeably different across the two papers. It would be helpful to understand what differences in training or evaluation setup might explain these shifts.

---

> > > ### Author Response · Authors · 2026-04-04
> > >
> > > We sincerely thank the reviewer for the constructive feedback and the positive reassessment after the rebuttal.
> > >
> > > Thank you for raising CDM. We agree it is a relevant recent CTKD baseline, especially given its focus on overlapping teacher-student pairs and instruction-following benchmarks. CDM improves CTKD by refining both sequence-level and vocabulary-level correspondence, specifically through entropy-weighted DTW for sequence alignment and contextual dynamic Top-K vocabulary mapping. It is therefore most closely related to the explicit vocabulary-alignment family represented in our paper by DSKD and MinED, rather than to the vocabulary-agnostic direction taken by E-SCOT.
> > >
> > > The main reason we did not include CDM in the experimental table is that we did not have a fully matched same-protocol reproduction, and we wanted to avoid an unfair head-to-head comparison. In particular, the published CDM setup differs from ours in several important respects: it uses a 3-epoch SFT + 7-epoch continual distillation schedule, a different hardware platform (8 Ascend 910B), and its released instruction-following evaluation script runs with a single seed, whereas our paper uses a unified 10-epoch pipeline with 5-seed averaging on 8 A100 (80GB) GPUs. In addition, our paper uses full fine-tuning for smaller pairs and LoRA for larger pairs, whereas the published/released CDM setup does not match this same adaptation protocol.
> > >
> > > We therefore believe that the observed absolute-score differences in the overlapping settings are more plausibly explained by differences in training/evaluation protocol than by a single method effect. For reference, under our pipeline, the reported average instruction-following scores on the overlapping teacher-student settings are generally higher than those reported in CDM, though we emphasize that this comparison should be treated as indicative rather than conclusive given the protocol differences noted above.
> > >
> > > As for competitiveness, we agree that CDM is a meaningful and reasonably strong baseline. Based on its design principles, E-SCOT should remain competitive in low-overlap settings under a fully matched pipeline, as its span-anchored mechanism inherently avoids noise from explicit vocabulary mapping.
> > >
> > > We have already cited CDM in the current draft, but we agree that the discussion is currently too limited; in the revision, we will expand the related work section accordingly. We thank the reviewer again for this helpful discussion and constructive feedback.

---

### Official Review · Reviewer_cMMk · 2026-03-13

**Soundness:** 3
**Presentation:** 3
**Significance:** 3
**Originality:** 3
**Overall Recommendation:** 4
**Confidence:** 4

**Summary:**

This paper proposes a knowledge distillation method focused on the mismatched student/teacher tokenizer setting, specifically when there is low-vocabulary overlap setting between source tokenizer and target tokenizer. This method comprises of a 1) vocabulary agnostic ground metric (1D wasserstein), 2) a sparse structural alignment-based token correspondence method to compute teacher support sets for student tokens, and a Renyi-entropy based reweighting scheme to place higher learning weight on token pairs with higher distribution mismatch. Results on CTKD settings show improved performance compared to other CTKD baselines, and an ablation study demonstrates the contribution of each component of the method.

**Compliance With Llm Reviewing Policy:**

Affirmed.

**Final Justification:**

The rebuttal addressed some key concerns with the extent of claims, promising to adjust them in the final version of the paper. Key clarity points were also addressed, which will also be updated in the final version. I adjusted my evaluation accordingly.

**Key Questions For Authors:**

1. How were the hyperparameters ($\tau$, $\rho$, $\lambda$) selected? This can help improve the reproducibility of this work.
2. What precisely is the fine-tuning sequence for the teacher and then the student? Is the teacher fine-tuned on the task for 10 epochs as well?
3. How does the support set construction account for white space? Is it always the case that corresponding tokens will overlap in their span positions? -- I am wondering about the case represented by this toy example of "a b c d" and "abcd" where the first string is 7 characters, and the second is 4.

**Limitations:**

yes

**Strengths And Weaknesses:**

Strengths:
1. The method's reweighting of token importance is well motivated, intuitive, and shown in the ablation analysis to improve the overall performance of the proposed method.
2. The method is extensively baselined against similar cross-tokenizer KD methods and shown to outperform them on a number of benchmarks, including the important SFT baseline.
3. The method has no actual expensive OT step, and can still outperform OT based CTKD methods, which might pave a way forward for avoiding this expensive step during KD.

Weaknesses
1. The vocabulary agnostic ground metric proposed as part of the method is not a novel contribution as it is the same metric from prior work (ULD). While it is fine to reuse the metric, the contribution of the paper should exclude this metric.
2. The memory and compute cost overhead is not discussed in this work; despite the nice reweighting idea, the reweighting is comprised of several entropy computations across varying Renyi alphas and across the vocabulary, combined into the student token level given the teacher neighborhood, and then normalized across matched positions. This step seems computationally expensive, and further analysis into the FLOPs required and memory needed can strengthen the applicability of this approach.
3. Despite the paper's narrative focusing on a low overlap negative transfer regime, the proposed method E-SCOT is not specifically shown to be particularly better in this setting. Phi->Qwen has low overlap, but other methods like ULD also outperform the SFT baseline.

---

> ### Author Rebuttal · Authors · 2026-03-31
>
> Thank you for the careful reading. We address your concerns below.
>
> 1. **Novelty of the vocabulary-agnostic ground metric.**
> We agree that the vocabulary-agnostic ground metric is not a novel component of this paper; it is adopted from ULD as a principled way to compare teacher and student distributions without a shared vocabulary. Our contribution lies in how this metric is integrated into a new CTKD framework: (i) a span-anchored lexical alignment that constructs a sparse, locality-preserving support set from character-span overlap, and (ii) a Renyi-entropy adaptive reweighting mechanism that prioritizes positions with larger uncertainty-profile discrepancy. We will revise the paper to make this contribution boundary explicit and avoid overstating novelty on the metric itself.
>
> 2. **Compute/memory overhead.**
> Although E-SCOT avoids dense sequence-level OT optimization, it still incurs additional cost from vocabulary-level ULD comparison and entropy-based token reweighting. We therefore added an empirical cost analysis under the same hardware and training setup in the Llama3-8B -> Gemma2-2B setting:
>
> | Method | Peak Mem Alloc (GB) | Step Time (s, max across ranks) | Throughput (tok/s) |
> |---|---:|---:|---:|
> | SFT | 11.39 | 0.94 | 2570.99 |
> | DSKD | 42.01 | 1.35 | 1784.03 |
> | MultiLevelOT | 55.21 | 2.12 | 1156.08 |
> | ULD | 42.99 | 2.57 | 1079.15 |
> | E-SCOT | 41.57 | 2.31 | 1041.82 |
> | MinED | 39.08 | 2.64 | 903.64 |
>
> E-SCOT uses less peak memory than ULD and substantially less than MultiLevelOT, while remaining in the same runtime range as ULD/MinED. As expected, all CTKD methods are more expensive than plain SFT, but these results show that E-SCOT does not introduce prohibitive practical overhead relative to prior CTKD methods. We will also revise the paper so that "linear-time" refers specifically to support-set construction rather than end-to-end training cost.
>
> 3. **Whether E-SCOT is truly stronger in the low-overlap regime.**
> We agree that the intended claim should be stated more precisely. Our point is not that every prior CTKD method fails in every low-overlap setting. In particular, ULD can remain relatively robust compared with alignment-based methods such as DSKD and MinED when vocabulary overlap is low. However, ULD remains strictly token-level and does not explicitly model local sequential consistency, which limits its performance. This directly motivated our design. E-SCOT retains the vocabulary-agnostic advantage of ULD, while adding span-anchored lexical alignment and Renyi-entropy adaptive reweighting to better exploit local structure and allocate distillation budget to informative positions. We will sharpen the wording accordingly: alignment-based CTKD methods such as DSKD and MinED are more prone to negative transfer in low-overlap regimes, ULD is relatively more robust but structurally limited, and E-SCOT combines vocabulary-agnostic matching with stronger sequence-aware supervision.
>
> 4. **Hyperparameter selection.**
> We apologize if this was not sufficiently visible. The key smoothing parameter $\lambda$ was selected via validation-set sensitivity analysis and then fixed for all main experiments. As shown in Fig. 5, $\lambda = 0.25$ provides a good trade-off between stability and adaptivity. The distillation temperature $\tau$ and KD rate $\rho$ are not unique to our method; for fair comparison we follow DSKD and use $\tau = 2$ and $\rho = 0.5$. Teacher/student optimization hyperparameters are listed in the appendix, and the student configuration is kept consistent between the SFT baseline and distillation runs. We will summarize these points more explicitly in the paper.
>
> 5. **Teacher/student fine-tuning sequence.**
> The sequence is: first fine-tune the teacher on task data for 10 epochs; then freeze the teacher; then train the student either with standard SFT or with a CTKD objective using the frozen teacher. We will clarify this workflow in the revision.
>
> 6. **Whitespace and support-set construction.**
> Our support set is defined by character-span overlap on the same underlying raw text, not by token identity or one-to-one token correspondence. Concretely, we tokenize the same raw string with both tokenizers, obtain offset mappings, and connect tokens whose spans overlap. For example, for the shared raw text "a b c d", one tokenizer may produce ["a", " b", " c", " d"] with spans [0,1), [1,3), [3,5), [5,7), while another produces ["a", "b", "c", "d"] with spans [0,1), [2,3), [4,5), [6,7). The resulting mapping is then "a" -> " a", "b" -> " b", "c" -> " c", and "d" -> " d" based on span overlap. If the example instead compares two different raw strings (e.g., "a b c d" vs. "abcd"), then they do not share the same character coordinate system, which is outside the assumption of our method. We will clarify this assumption and add a brief example in the revision.
>
> We appreciate these comments, which help us sharpen both the precision of our claims and the clarity of the presentation.

---

> > ### Author Rebuttal · Reviewer_cMMk · 2026-04-02
> >
> > My key concerns in the review were mostly related to claim scoping and clarity on some portions. I am adjusting my score to reflect the resolution of some weaknesses which will be addressed in the writing in the paper as stated by the authors. The new score reflects the improved presentation/originality (due to claim adjustment) after reading the response. Thanks for being receptive to some key feedback!

---

> > > ### Author Response · Authors · 2026-04-03
> > >
> > > Dear Reviewer cMMk,
> > >
> > > Thank you again for your careful review and constructive feedback. We sincerely appreciate your updated evaluation of our work and are glad that our revisions helped address your concerns.
> > >
> > > Sincerely,
> > > The Authors

---

### Official Review · Reviewer_nVM5 · 2026-03-13

**Soundness:** 3
**Presentation:** 3
**Significance:** 3
**Originality:** 2
**Overall Recommendation:** 4
**Confidence:** 2

**Summary:**

This paper investigates a key issue in Cross-Tokenizer Knowledge Distillation (CTKD): when the vocabulary overlap between the teacher model and the student model is low, the performance of existing CTKD methods is even worse than that of simple SFT baseline. The authors define this phenomenon as the "Low-Overlap Negative Transfer Regime". The paper propose the E-SCOT framework, which includes: a Vocabulary-Agnostic Ground Metric based on 1D Wasserstein-1 distance, which measures the difference in teacher-student distribution by comparing the shape of the sorted probability distributions, without requiring a shared vocabulary index; Span-Anchored Lexical Alignment based on character span overlap, which uses a two-pointer scan to construct a sparse matching matrix in linear time, avoiding the quadratic complexity of dense optimal transport; and Renyi-Entropy Adaptive Reweighting, which dynamically allocates the distillation budget through multi-order normalized Renyi entropy differences, making the model focus more on locations with significant teacher-student uncertainty differences. Experiments were conducted on multiple teacher-student model pairs with different vocabulary overlap rates. The results show that E-SCOT outperforms existing baselines in all settings, especially effectively eliminating negative transfer in low-overlap scenarios.

**Compliance With Llm Reviewing Policy:**

Affirmed.

**Final Justification:**

Fully resolved

**Key Questions For Authors:**

1. Regarding the generalization of experimental scale: All experiments were conducted on Dolly-15K. On larger training datasets, is the low-overlap negative transfer still significant? Does the advantage of E-SCOT over the baseline diminish with increasing data size? Providing at least one set of large-scale data experiments would significantly enhance the paper's persuasiveness.
2. Regarding the specificity of Rényi reweighting: Rényi-entropy adaptive reweighting is essentially a general token-level loss reweighting strategy. Have the authors verified the effectiveness of this strategy under the same vocabulary KD setting? If the strategy is equally effective under the same vocabulary setting, its contribution is not unique to CTKD, which would affect the understanding of its role in this framework.
3. Regarding computational efficiency: The paper emphasizes the advantage of E-SCOT's linear time complexity but does not provide actual wall-clock time or GPU memory comparisons. Could you provide a comparison of training time and peak memory usage between E-SCOT and ULD, MultiLevelOT, and DSKD on the same hardware? This is crucial for evaluating the practicality of the methods.

**Limitations:**

Yes

**Strengths And Weaknesses:**

Strengths And Weaknesses:

Strengths
1. This paper clearly identifies and quantifies the phenomenon of low-overlap negative transfer intervals. Figure 2 visually demonstrates, through systematic experiments, the performance changes of existing CTKD methods (MinED, DSKD, ULD) relative to the SFT baseline at different vocabulary overlap rates, convincingly illustrating that most existing methods exhibit negative transfer when the vocabulary overlap rate is below approximately 35%. This finding has significant practical implications for the CTKD field, as cross-model family distillation is becoming increasingly common in practical deployments.
2. The experiments covered four main teacher-student model pairs, with vocabulary overlap rates ranging from 15.2% to 75.6%, involving multiple model families including Phi-3, Llama-3, Mistral, Qwen2, Gemma-2, OPT, and TinyLlama. Ablation experiments (Table 2) systematically validated the contributions of each component, and hyperparameter sensitivity analyses were also relatively complete.
3. E-SCOT relies solely on deterministic character span alignment and reweighting coefficients for stopping gradients, without introducing additional model parameters, which has engineering advantages in real-world large-scale training.

Weaknesses
1. All experiments were trained using only the relatively small instruction fine-tuning dataset Dolly-15K. Does the low-overlap negative transfer phenomenon still exist on larger training datasets (e.g., hundreds of thousands or millions of samples)? Does the advantage of E-SCOT remain significant? The lack of discussion on data scale weakens the generalizability of the conclusions.
2. The evaluation metric used only is Rouge-L, a metric based on surface n-gram overlap, which is insufficient to comprehensively reflect generation quality. Results based on LLM-as-judge or human evaluation are lacking, especially for open generation tasks such as Vicuna-Eval.
3. In Table 2, the performance of "w/o Span Constraint" is 26.51, a difference of 0.40 compared to the full E-SCOT (26.91), while the performance of "w/o Adaptivity" is 26.00, a difference of 0.91. This indicates that Rényi reweighting contributes more than Span alignment, but this is not discussed in depth in the paper. In particular, Rényi reweighting is essentially a general token-level loss reweighting strategy; can it also improve the performance of CTKD with the same vocabulary? If so, its contribution is not unique to CTKD.
4. The poor performance of MultiLevelOT in experiments (even below ULD and SFT in multiple settings) lacks in-depth analysis. Is this due to limitations of the method itself, or insufficient hyperparameter tuning?
5. The paper lacks experiments conducted on newer LLMs, such as those by Qwen3 et al.

---

> ### Author Rebuttal · Authors · 2026-03-31
>
> Thank you for the positive assessment and constructive comments. We appreciate that you recognized both the practical importance of the low-overlap regime and the engineering simplicity of our design. Our point-by-point responses are below.
>
> 1. **Use of Dolly-15K only.**
> We agree that relying only on Dolly-15K limits the generalizability of the current submission, and that larger-scale evidence is important. To examine generalization beyond Dolly-15K, we added an experiment on an UltraChat-200K subset in a larger-scale setting: Phi-4-14B -> Qwen3-4B, max sequence length 1024, trained for 3 epochs (tokenizer overlap ratio: 64.22%). We also broadened evaluation beyond the original instruction-following setup by testing PiQA, ARC-C, MMLU, and AGI-CN:
>
> | Phi4-14B -> Qwen3-4B | PiQA | ARC-C | MMLU | AGI-CN | Avg |
> |---|---:|---:|---:|---:|---:|
> | teacher | 0.8128 | 0.5606 | 0.7691 | 0.5750 | 0.6794 |
> | Base | 0.7514 | 0.5410 | 0.6832 | 0.5482 | 0.6310 |
> | SFT | 0.7617 | 0.5546 | 0.6763 | 0.6067 | 0.6498 |
> | DSKD | 0.7622 | 0.5427 | 0.6736 | 0.5935 | 0.6430 |
> | E-SCOT | 0.7769 | 0.5606 | 0.6800 | 0.6227 | 0.6601 |
>
> In this larger-data, larger-model, fewer-epoch, longer-context setting, E-SCOT still outperforms both SFT and DSKD. Importantly, DSKD remains below SFT on the overall average (0.6430 vs. 0.6498), indicating that negative transfer can still arise beyond the original Dolly-15K setup. We believe it provides meaningful new evidence beyond the original controlled setting.
>
> 2. **Relying only on Rouge-L.**
> We agree that Rouge-L alone is insufficient, especially for open-ended instruction-following tasks. To address this, we added an LLM-as-Judge evaluation on Vicuna-Eval for the Phi-3-mini-3.8B -> Qwen2-0.5B setting, using Qwen2.5-Plus (qwen-plus-2025) as the judge model:
>
> | Comparison | Win Rate | Breakdown (Win/Tie/Loss) |
> |---|---:|---:|
> | vs SFT | 64.38% | 49 / 5 / 26 |
> | vs MinED | 66.25% | 48 / 10 / 22 |
> | vs ULD | 58.13% | 41 / 11 / 28 |
> | vs DSKD | 69.38% | 51 / 9 / 20 |
>
> E-SCOT wins against all four baselines, showing that its advantage is not limited to Rouge-L and is also supported by judge-based evaluation on an open-generation benchmark. We believe that this result can provide substantially stronger evidence than Rouge-L alone.
>
> 3. **Role and specificity of Renyi reweighting.**
> We do not claim that Renyi-based reweighting is useful only in CTKD. To clarify this point, we additionally tested it in a same-vocabulary KD setting (Llama2-7B -> TinyLlama-1.1B). Compared with the skewed reverse KL baseline, adding Renyi reweighting improves performance on all three benchmarks:
>
> | Method | S-NI | UnIn | Vicuna |
> |---|---:|---:|---:|
> | Skewed Reverse KL | 31.19 | 30.43 | 17.01 |
> | + Renyi reweighting | 32.15 | 31.65 | 17.06 |
>
> These results show that Renyi reweighting is useful for same-vocabulary KD as well, and we believe it plays a more critical role in cross-tokenizer CTKD, especially in low-overlap settings where supervision quality is more heterogeneous and noisy across positions. In this sense, it is especially effective when combined with sparse span-constrained support. We will clarify this scope more explicitly in the revision.
>
>
> 4. **Weak performance of MultiLevelOT.**
> We agree that this point deserves clearer discussion. Relative to ULD, MultiLevelOT introduces sequence-aware token-level OT, sequence-level Sinkhorn matching, and top-k truncation to capture richer global structure and improve cross-tokenizer alignment. However, these choices also make the method more dependent on reliable global correspondences and introduce multiple method-specific hyperparameters. In our experiments, we followed the original paper's parameter settings.
>
> Our interpretation is that this issue is particularly pronounced in low-overlap cross-tokenizer settings: when many teacher-student correspondences are already weak or ambiguous, the more global transport mechanisms in MultiLevelOT can become more sensitive to alignment noise and may spread supervision mass over unreliable matches, thereby weakening token-level guidance. The relatively larger hyperparameter space may further increase tuning sensitivity. In contrast, E-SCOT restricts supervision to a deterministic, locality-preserving support set and reweights only within this local neighborhood, making the alignment mechanism more direct and robust in low-overlap settings. We will add this discussion in the revision.
>
> 5. **Newer models.**
> We have also added an additional experiment with a newer model pair, Phi-4-14B -> Qwen3-4B, to further test the generality of E-SCOT beyond the original set of teacher-student families.

---

> > ### Author Rebuttal · Reviewer_nVM5 · 2026-04-03
> >
> > NA

---

> > > ### Author Response · Authors · 2026-04-03
> > >
> > > Dear Reviewer nVM5,
> > >
> > > Thank you for your time and for acknowledging that our response has addressed your concerns. We appreciate your careful evaluation.
> > >
> > > Sincerely,
> > > The Authors

---

### Official Review · Reviewer_Qk3h · 2026-03-23

**Soundness:** 3
**Presentation:** 2
**Significance:** 2
**Originality:** 3
**Overall Recommendation:** 4
**Confidence:** 3

**Summary:**

The paper proposes Entropy-aware Span-Constrained Optimal Transport (E-SCOT) to address the challenge of Cross-Tokenizer Knowledge Distillation (CTKD) in Large Language Models (LLMs). The authors first highlight a "Low-Overlap negative transfer regime," demonstrating empirically that existing CTKD methods (e.g., DSKD, MinED) that rely on explicit token or hidden-state alignment often underperform a standard Supervised Fine-Tuning (SFT) baseline when the vocabulary overlap between the teacher and student models is small. To overcome this, E-SCOT avoids explicit alignment learning and quadratic sequence-level Optimal Transport (OT) costs. The method comprises three key components:

1. A vocabulary-agnostic ground metric that computes a 1D Wasserstein distance on sorted probability profiles (adapted from ULD).

2. A span-anchored lexical alignment that uses character-level offset overlaps in the raw text to construct a sparse, deterministic matching matrix in linear time $\mathcal{O}(|\mathcal{I}| + |\mathcal{J}|)$.

3. A Rényi-entropy adaptive reweighting scheme that dynamically allocates a larger distillation budget to token pairs exhibiting significant uncertainty-profile gaps (using multi-order Rényi entropy).

Extensive experiments on instruction-following datasets across diverse teacher-student model pairs (e.g., Llama-3 to Gemma-2, Phi-3 to Qwen-2) show that E-SCOT consistently outperforms existing CTKD baselines and avoids negative transfer in low-overlap regimes.

**Compliance With Llm Reviewing Policy:**

Affirmed.

**Final Justification:**

The rebuttal addressed my concerns; maintaining my Weak Accept score.

**Key Questions For Authors:**

1. Computational Overhead: While the sequence alignment is linear, the ULD Wasserstein metric requires sorting the full probability distribution, adding an $\mathcal{O}(|V| \log |V|)$ cost per token. For vocabularies of 128K or 256K, this is computationally expensive. How does the actual empirical training throughput (e.g., tokens/sec) of E-SCOT compare to SFT and baseline CTKD methods?

2. Data Scale: Instruction tuning and distillation typically leverage much larger datasets than Dolly-15k to avoid overfitting, especially when training for 10 epochs. Have you tested E-SCOT on larger datasets, and does the "Low-Overlap negative transfer" phenomenon still hold when models are trained on 100k+ examples for 1-2 epochs?

3. Value of the parameter lambda and its generalizability across overlap regimes:The authors appear to have fixed $\lambda$ to 0.25 globally across all of their main experiments based on the Dolly validation set. However, the core premise of the paper is the "Low-Overlap negative transfer regime". It stands to reason that a highly mismatched teacher-student pair (e.g., Phi-3 to Qwen2, 15.2% overlap) might exhibit vastly different uncertainty-profile gaps than a high-overlap pair (e.g., Mistral to TinyLlama, 75.6% overlap). The paper does not explore whether the optimal $\lambda$ shifts depending on the severity of the vocabulary mismatch. Addressing whether $\lambda$ should be dynamically scaled based on the inherent vocabulary overlap ratio would make this section perfectly bulletproof.

**Limitations:**

No. The authors included a generic "Impact Statement" but entirely failed to discuss the technical limitations of their work.Constructive suggestions for improvement: Please include a dedicated "Limitations" section discussing:

1. The computational overhead of sorting vocabulary-sized logits for the Wasserstein metric, which adds a significant $\mathcal{O}(|V| \log |V|)$ cost per token, especially for modern large-vocabulary tokenizers.

2. The reliance on exact character span alignment, which might be brittle or require complex engineering workarounds if tokenizers apply different irreversible string normalizations (e.g., Unicode normalization) or handle byte-level fallbacks differently.

**Strengths And Weaknesses:**

Strengths:

Originality & Pragmatism (Alignment Mechanism): The proposed span-anchored lexical alignment is an intuitive, simple, and highly efficient method to establish token correspondences across disparate tokenizers. By leveraging shared raw character spans, the approach inherently preserves linguistic locality and strictly enforces monotonicity, successfully bypassing the complex and computationally prohibitive $\mathcal{O}(N^2)$ iterative Sinkhorn OT or DTW algorithms used in prior works (like MultiLevelOT or MinED).

Originality (Adaptive Reweighting): The integration of the normalized Rényi entropy spectrum to measure uncertainty-profile gaps is a highly creative and theoretically sound approach. Since multi-order Rényi entropy can capture structural and geometric differences in predictive distributions that standard Shannon entropy might miss (as elegantly illustrated in the Figure 4 toy experiment), it serves as a principled metric to identify "informative" tokens for distillation.

Significance: The problem of cross-tokenizer distillation is a major bottleneck for the open-source LLM community, especially as newer models are released with vastly different and larger vocabularies (e.g., Llama-3 with 128k, Qwen2.5 with 152k, Gemma-2 with 256k). A stable, robust CTKD method that avoids negative transfer offers substantial practical utility.

Weaknesses:

Soundness (Computational Complexity Claims): The authors repeatedly claim their method is highly efficient and operates in linear time because the sequence alignment step takes $\mathcal{O}(|\mathcal{I}| + |\mathcal{J}|)$. However, calculating the vocabulary-agnostic ground metric (Equation 3) requires sorting the entire vocabulary-sized probability distribution for every aligned token pair. For modern models like Gemma-2 with a 256K vocabulary, this $\mathcal{O}(|V| \log |V|)$ sorting operation introduces massive computational overhead and memory bandwidth bottlenecks on GPUs. The paper completely omits any discussion or empirical measurement of the actual training throughput (e.g., tokens/second or wall-clock time) compared to standard SFT or projection-based CTKD methods.

Soundness (Experimental Scale & Overfitting): The distillation experiments are limited to training on the Dolly-15k dataset for 10 epochs. This setup is highly prone to severe overfitting. High ROUGE-L scores after 10 epochs on 14k samples may heavily reflect overfitting to specific phrasing styles of the dataset rather than generalized capability transfer. Modern LLM distillation generally requires larger, higher-quality datasets (e.g., 100k+ samples) for fewer epochs (1-3).

Soundness (Heuristic Span Weighting): In Equation 5, the matching matrix assigns uniform weights to all overlapping teacher tokens. If a student token overlaps with one teacher token by 5 characters and another by just 1 character, weighting their supervision signals equally seems theoretically suboptimal. A proportional overlap weighting would be more mathematically principled.

---

> ### Author Rebuttal · Authors · 2026-03-31
>
> Thank you for the constructive and detailed feedback. Below we address each concern point by point.
>
> 1. **Computational complexity.**
> We agree that "linear-time" should be stated more precisely: the linear complexity applies to the sequence-alignment step, i.e., constructing the sparse support set via a two-pointer sweep over sorted token spans. The overall training cost also includes the vocabulary-level distribution comparison used in the ULD ground metric, which requires sorting vocabulary-sized distributions. To address this concern directly, we added an empirical cost analysis under the same hardware and training setup in the Llama-3-8B -> Gemma2-2B setting:
>
> | Method | Peak Mem Alloc (GB) | Step Time (s, max across ranks) | Throughput (tok/s) |
> |---|---:|---:|---:|
> | SFT | 11.39 | 0.94 | 2570.99 |
> | DSKD | 42.01 | 1.35 | 1784.03 |
> | MultiLevelOT | 55.21 | 2.12 | 1156.08 |
> | ULD | 42.99 | 2.57 | 1079.15 |
> | E-SCOT | 41.57 | 2.31 | 1041.82 |
> | MinED | 39.08 | 2.64 | 903.64 |
>
> These results show that, while all CTKD methods are more expensive than plain SFT, E-SCOT does not introduce prohibitive practical overhead relative to prior CTKD baselines. In particular, E-SCOT uses substantially less peak memory than ULD and MultiLevelOT, slightly less than DSKD, and achieves runtime/throughput in the same range as existing CTKD methods. We will revise the paper accordingly so that "linear-time" refers specifically to support-set construction rather than end-to-end training cost.
>
> 2. **Data scale and possible overfitting.**
> We agree Dolly-15K is relatively small for modern LLM distillation. To examine generalization beyond Dolly-15K, we added a larger-scale experiment on an UltraChat-200K subset: Phi-4-14B -> Qwen3-4B, max sequence length 1024, for 3 epochs (tokenizer overlap ratio: 64.22%). We also broadened the evaluation beyond instruction-following by testing PiQA, ARC-C, MMLU, and AGI-CN:
>
> | Phi4-14B -> Qwen3-4B | PiQA | ARC-C | MMLU | AGI-CN | Avg |
> |---|---:|---:|---:|---:|---:|
> | teacher | 0.8128 | 0.5606 | 0.7691 | 0.5750 | 0.6794 |
> | Base | 0.7514 | 0.5410 | 0.6832 | 0.5482 | 0.6310 |
> | SFT | 0.7617 | 0.5546 | 0.6763 | 0.6067 | 0.6498 |
> | DSKD | 0.7622 | 0.5427 | 0.6736 | 0.5935 | 0.6430 |
> | E-SCOT | 0.7769 | 0.5606 | 0.6800 | 0.6227 | 0.6601 |
>
> In this larger-data, larger-model, fewer-epoch, longer-context setting, E-SCOT still outperforms both SFT and DSKD, and DSKD remains below SFT on average (0.6430 vs. 0.6498), indicating that negative transfer can still arise beyond the Dolly-15K setup.
>
> 3. **Uniform weighting within a span neighborhood.**
> Our use of uniform weighting in Eq. 5 was motivated by simplicity and stability. The current ablation already suggests that the main gain comes from enforcing span locality rather than from a sophisticated weighting rule inside the neighborhood: "Span-Anchored" and "Span (Max)" are very close. We will clarify this interpretation and note that proportional-overlap weighting is a promising extension.
>
> 4. **Choice and generalizability of $\lambda$.**
> We selected $\lambda$ via validation-set sensitivity analysis and fixed it across the main experiments to avoid pair-specific tuning. We agree, however, that the optimal $\lambda$ can depend on the vocabulary-overlap regime. We now add supporting evidence for this point: on the high-overlap Mistral-7B -> TinyLlama setting (75.6%), $\lambda$ <= 0.25 consistently outperforms $\lambda$ >= 0.5, with $\lambda$ = 0.1 performing best on Vicuna-Eval and Unnatural Instructions, while $\lambda$ = 0.25 performs best on SuperNI:
>
> | $\lambda$ | Vicuna-Eval | SuperNI | Unnatural Instructions |
> |---|---:|---:|---:|
> | 0.1 | 17.72 | 31.93 | 32.39 |
> | 0.25 | 17.54 | 32.28 | 32.16 |
> | 0.5 | 16.97 | 30.76 | 30.87 |
> | 0.75 | 17.07 | 29.64 | 30.43 |
>
> For the low-overlap Phi-3 -> Qwen2 setting (15.2%), our sensitivity study indicates that $\lambda$ = 0.25 is strongest. These results suggest that overlap-aware tuning of $\lambda$ is promising. At the same time, a single fixed $\lambda$ = 0.25 already performs robustly across a broad overlap range (15.2%-75.6%), which is why we used it as the default in the main experiments. We will revise the paper to make both points explicit.
>
> 5. **Limitations.**
> We will add a dedicated limitations discussion covering: (i) the nontrivial cost of sorting vocabulary-sized distributions, (ii) possible brittleness under tokenizer normalization differences, and (iii) the current validation still being limited to moderate model sizes and context lengths.
>
> Thank you again for the detailed suggestions. We believe these clarifications and additions will make the paper more accurate and substantially stronger.

---

> > ### Author Rebuttal · Reviewer_Qk3h · 2026-04-02
> >
> > Thank you for the thorough and constructive rebuttal. You have directly addressed my primary concerns:
> >
> > 1. Computational Overhead: The new cost analysis table clarifies the practical overhead, showing that E-SCOT remains competitive in terms of tokens/sec.
> >
> > 2. Data Scale: The additional experiments on the UltraChat-200K subset effectively alleviate my worries about overfitting on Dolly-15k.
> >
> > 3. Lambda & Limitations: I appreciate the extra sensitivity analysis for the lambda parameter across different overlap regimes, as well as your commitment to adding a dedicated limitations section.
> >
> > I'll maintain my score of Weak Accept.

---

> > > ### Author Response · Authors · 2026-04-03
> > >
> > > Dear Reviewer Qk3h,
> > >
> > > Thank you for your thorough and constructive feedback. We sincerely appreciate your positive evaluation and are glad that our additional analyses helped address your concerns. We will incorporate these clarifications into the final version of the paper.
> > >
> > > Sincerely,
> > > The Authors

---

### Decision · Program_Chairs · 2026-04-30

**Decision:**

Accept (regular)

**Comment:**

This paper proposes a linear-time cross-tokenizer knowledge distillation framework that mitigates negative transfer in low-vocabulary-overlap scenarios through sparse optimal transport and entropy-adaptive reweighting, achieving state-of-the-art performance. Reviewers initially raised concerns regarding computational overhead, limited data scale, the absence of recent large language model baselines, and insufficient insights into hyperparameter selection. After the rebuttal, these issues were satisfactorily addressed. The remaining concern is limited novelty, as the contribution primarily stems from a well-executed integration of existing ideas. Nevertheless, all reviewers supported the paper due to the problem's significance, its intuitive analysis of the low-overlap negative transfer phenomenon, and its well-motivated, technically sound methodology. Therefore, this paper is recommended for acceptance.